# Tuning ferroelectric phase transition temperature by enantiomer fraction

Chang-Chun Fan [1,2], Cheng-Dong Liu[1,2], Bei-Dou Liang [1], Wei Wang [1], Ming-Liang Jin[1], Chao-Yang Chai [1], Chang-Qing Jing[1], Tong-Yu Ju[1], Xiang-Bin Han [1] ✉ & Wen Zhang [1] ✉

Tuning phase transition temperature is one of the central issues in phase transition materials. Herein, we report a case study of using enantiomer fraction engineering as a promising strategy to tune the Curie temperature ($T_C$) and related properties of ferroelectrics. A series of metal-halide perovskite ferroelectrics $(S-3AMP)_x(R-3AMP)_{1-x}PbBr_4$ was synthesized where 3AMP is the 3-(aminomethyl)piperidine divalent cation and enantiomer fraction x varies between 0 and 1 (0 and 1 = enantiomers; 0.5 = racemate). With the change of the enantiomer fraction, the $T_C$, second-harmonic generation intensity, degree of circular polarization of photoluminescence, and photoluminescence intensity of the materials have been tuned. Particularly, when x = 0.70 – 1, a continuously linear tuning of the $T_C$ is achieved, showing a tunable temperature range of about 73 K. This strategy provides an effective means and insights for regulating the phase transition temperature and chiroptical properties of functional materials.

Phase transition materials exhibit unique thermal, optical, electrical, mechanical, and magnetic properties in the process of phase transitions[1–6], and have important applications in memory[7–10], infrared detectors[11], integrated optoelectronic devices[12], and energy storage[13–16]. In particular, ferroelectric materials[17], as an important type of functional materials, possess high dielectric constant[18,19], large pyroelectric response[20–22], excellent piezoelectricity[23–26], nonlinear optical effects[27–29], and switchable photoresponses[30]. Since these properties and functions are directly related to ferroelectric phase transitions, controlling and tuning the phase transition temperature or Curie temperature ($T_C$) is the key to optimizing the performances of ferroelectric materials for various applications[31]. Common approaches to tuning the $T_C$ mainly focus on controlling the chemical compositions, including H/D isotope substitution[32–35], H/F substitution[36,37], metal/cation/anion doping[24,38], etc[39]. These approaches can alter the local intermolecular interactions and structural symmetries of the crystals and thereby affect the $T_C$[40]. However, these methods also have certain limitations. For example, H/D isotope substitution is only suitable for hydrogen-displacement type ferroelectricity[41]; and syntheses of fluorinated organic cations use some environmentally unfriendly

and highly toxic fluorinated reagents[42]. Furthermore, the H/D isotope and H/F substitutions correspond to either 0 or 1 doping, but linear doping of the two components for tuning the ferroelectricity and related properties is rarely explored and new phenomena may be missed.

In molecular ferroelectric crystals, the shape and size of the molecules greatly regulate the packing structures, which makes it possible to optimize the ferroelectricity through rational molecular design[43]. However, due to the highly sensitive nature of ferroelectricity to molecular structures and intermolecular interactions, sometimes, even substituting a single atom can result in the disappearance of ferroelectricity. Chirality has the advantage of having nearly the same intermolecular interactions with neighbors in the crystal structures, which can allow finely tuning of the $T_C$ by maintaining the ferroelectricity. Moreover, as a basic molecular structure feature, chirality has been exploited to design and construct chiral&ferroelectric materials to achieve bulk photovoltaic effect[44], chiroptical-coupled ferroelectric properties such as circularly polarized luminescence[45,46], and circularly polarized light (CPL) detection[47]. However, in these studies, little attention has been paid to the impact of enantiomeric excess on the

[1]Jiangsu Key Laboratory for Science and Applications of Molecular Ferroelectrics and School of Chemistry and Chemical Engineering, Southeast University, 211189 Nanjing, China. [2]These authors contributed equally: Chang-Chun Fan, Cheng-Dong Liu. ✉e-mail: hanxiangbin@seu.edu.cn; zhangwen@seu.edu.cn

structures and properties of the ferroelectrics, and the use of molecules with different chiral purities has never been considered when developing new materials. Retrospectively, research on enantiomer solid solutions mostly focuses on the difference in solubility and melting point of compounds with different enantiomer fractions of organic molecular crystals, which is generally used to guide chiral separation[48,49]. In organic chemistry, how to get enantiomerically pure molecules is always the focus because compounds with different enantiomeric purities often show different biological activities[50,51]. Generally, the potentials of materials with different enantiomer fractions have been overlooked in the discovery of new structures and properties as well as the cost consideration in material processing.

In this work, we report the utilization of enantiomer fraction engineering to achieve chiral doping to tune the $T_C$ and corresponding properties of molecular ferroelectrics. By controlling the enantiomeric excess of the chiral cation, a series of mixed chiral cation compounds $(S\text{-}3AMP)_x(R\text{-}3AMP)_{1-x}PbBr_4$ (3AMP = 3-(aminomethyl)piperidine divalent cation; enantiomer fraction $x = 0–1$) are obtained. Crystals **1Rac** ($x = 0.5$) and **1R/S** ($x = 0/1$) undergo ferroelectric phase transitions at 359 K and 432 K, respectively. By varying the enantiomer fraction, the $T_C$, second-harmonic generation (SHG) signal intensity, degree of circular polarization of photoluminescence, and photoluminescence intensity have been tuned. For conciseness, we mainly focus on **1S** ($x = 1$) but not **1R** ($x = 0$) to discuss the structures and properties.

## Results and discussion
### Ferroelectric property of 1Rac

Single crystals of the racemate **1Rac** and enantiomer **1S** were synthesized by a solution-evaporation method. The phase purity of the obtained crystals was verified by powder X-ray diffraction (PXRD) (Supplementary Fig. 1). Thermogravimetric analysis shows that crystals have good air and thermal stability below 610 K (Supplementary Fig. 2).

The step-like dielectric transition of the crystalline-powdered sample **1Rac** appears at 359 K with a dielectric change of about 30, indicating a phase transition (Fig. 1a). Differential scanning calorimetry (DSC) analysis shows reversible thermal peaks centered around the same temperature, corresponding to the phase transition (Fig. 1b). The variable temperature single crystal X-ray diffraction was performed to

determine the crystal structure of **1Rac** at 293 K (ferroelectric phase, FEP) and 370 K (paraelectric phase, PEP). **1Rac** crystallizes in the polar space group $Cc$ in the FEP (Fig. 1b and Supplementary Table 1). The non-centrosymmetric character was verified by a strong SHG signal comparable to KDP at 293 K (Supplementary Fig. 3). The asymmetric unit contains two 3AMP cations, two Pb ions, and eight Br ions (Supplementary Fig. 4). Hirshfeld surface analysis reveals a predominant role of the hydrogen bonds and additional van der Waals interactions between the cation and the Br anion (Supplementary Fig. 5). The ordered cations orient up and down alternatively between adjacent inorganic layers via N−H···Br hydrogen bonds (Supplementary Fig. 6 and Supplementary Table 2). At 370 K, the structure of **1Rac** corresponds to the PEP with the centrosymmetric space group $C2/c$. The asymmetric unit contains one 3AMP cation, one Pb ion, and five Br ions, of which 2 Br ions are located at special positions with 2 rotational symmetries and occupancy of 1/2 (Supplementary Fig. 3). Meanwhile, the 3AMP cations are located at 2-fold axes in a disordered state, resulting in a cancellation of the polarization (Fig. 1b). Therefore, the order-disorder transition of the 3AMP cations triggers the ferroelectric-paraelectric phase transition. The SHG signal switches off at around 370 K, i.e., the intensity decreases gradually below the $T_C$ and reaches almost zero above the $T_C$, revealing a polar symmetry change due to the FEP-PEP transition (Fig.1d). It can be seen from Supplementary Figs. 5 and 7 that the N−H·· Br interaction strength significantly weakens above the $T_C$, which accords with the order-disorder phase transition. Direct evidence of the ferroelectricity was confirmed by the well-shaped electric hysteresis loop by using the double-wave method performed on the crystal along the $c$-axis (Fig. 1e)[52]. A polarization of 0.50 μC·cm$^{-2}$ is measured with a coercive field of 7.0 kV/cm.

We previously reported the structure of the chiral ferroelectric **1S**[53]. Both **1R** and **1S** crystallize in the space group $P2_1$ (polar point group 2) in the FEP (Supplementary Fig. 8a). Cotton effect has been identified using circular dichroism spectra (Supplementary Fig. 8c). DSC and dielectric transition show that **1R/S** undergo a reversible ferroelectric-paraelectric phase transition at 432 K (Fig. 2a and Supplementary Fig. 8d). The PEP belongs to the SHG-inactive space group $P422$ (point group 422) (Supplementary Fig. 8b, e).

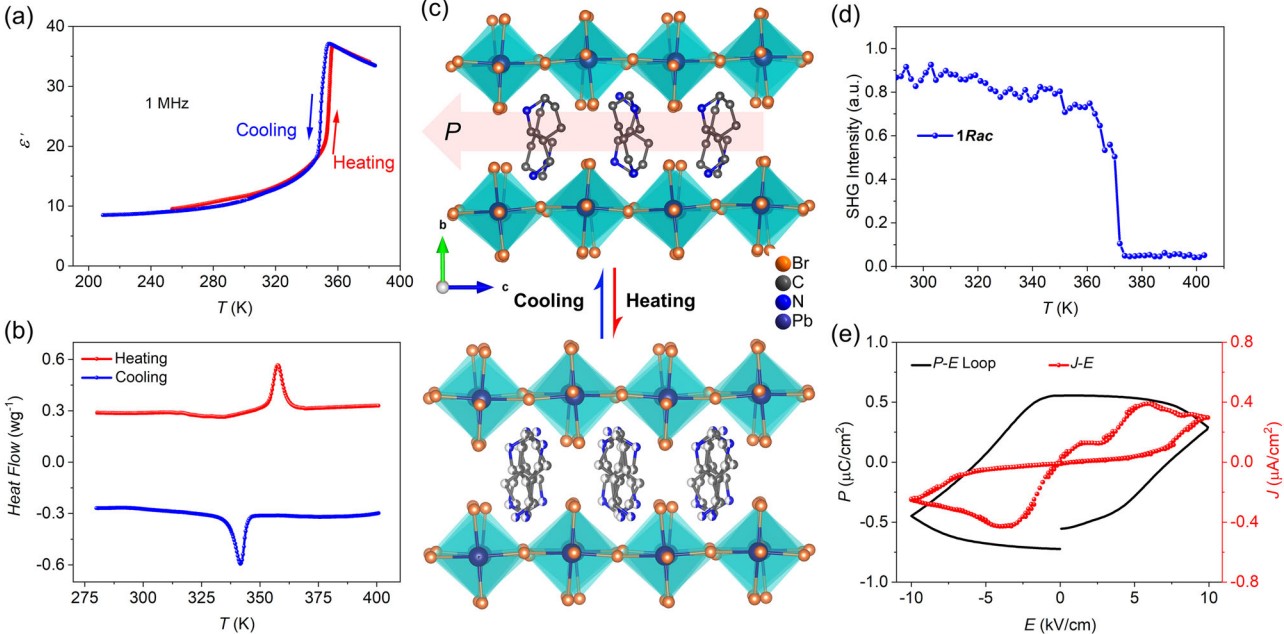

**Fig. 1 | Evidence of the ferroelectricity in 1Rac ($x = 0.5$). a** Dielectric transition; **b** DSC curves. **c** Crystal structures of **1Rac** in the ordered FEP (ferroelectric phase) and disordered PEP (paraelectric phase). Hydrogen atoms are omitted for clarity. **d** SHG signals. **e** P−E hysteresis loop recorded at 293 K in the FEP. Note: $P–E$ (polarization versus electric field curve) and $J–E$ (current density versus electric field curve).

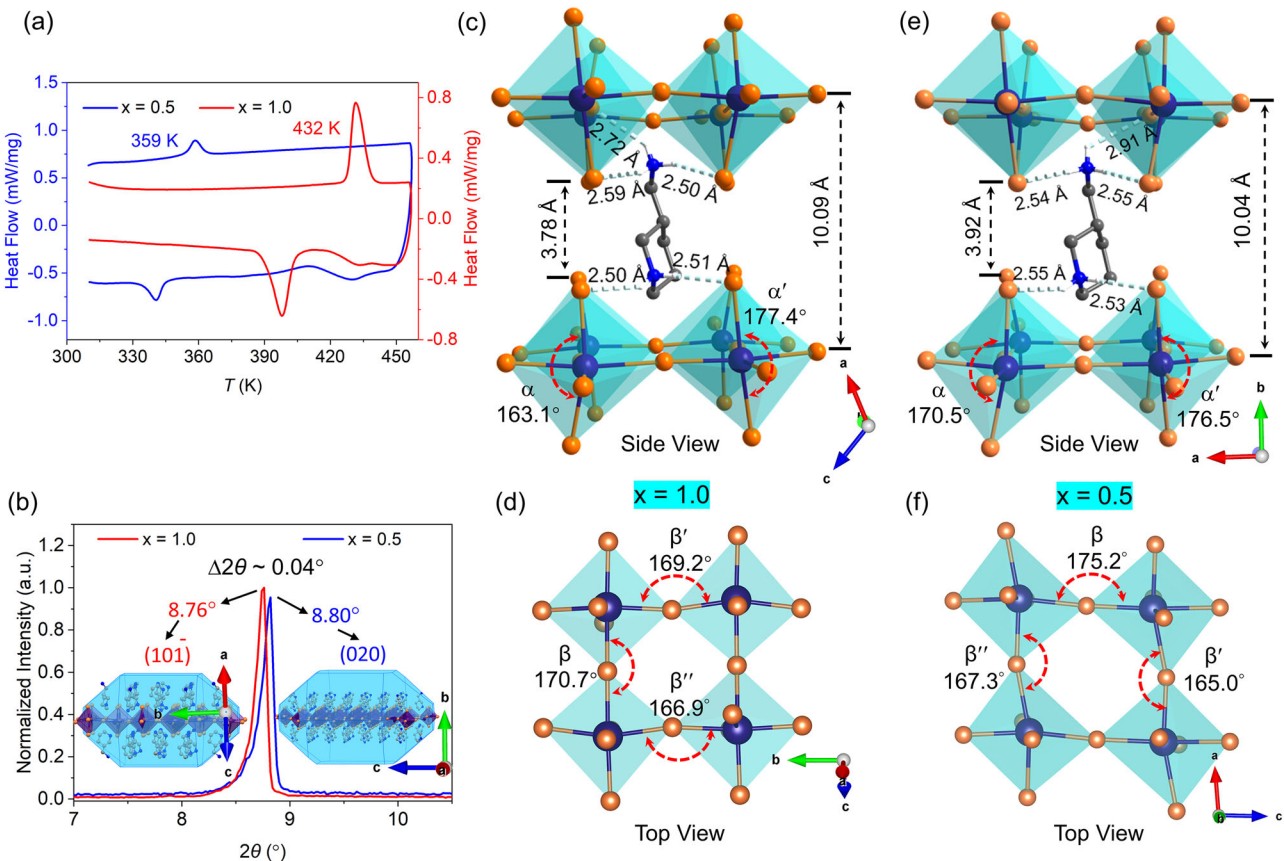

**Fig. 2 | Structural analysis and comparison of 1S and 1Rac (293 K). a** DSC curves of **1S** and **1Rac**. **b** PXRD peaks of **1S** and **1Rac**. **c**, **e** N−H···Br bonds and out-of-plane of the inorganic sublattice of **1S** and **1Rac**, respectively. **d**, **f** In-plane distortions of the inorganic sublattice of **1S** and **1Rac**, respectively.

## Structural analysis and comparison of 1R/S and 1Rac

It is found that the $T_C$ of **1S** is 73 K higher than the one of **1Rac** (Fig. 2a). To understand this difference, we compare and analyze their structures. The corresponding PXRD data show that from **1S** to **1Rac**, the peak representing the [PbBr₄] inorganic plane moves from 8.76° to 8.80° (Fig. 2b). According to Bragg's equation $2d\sin\theta = n\lambda$, the corresponding spacing d of adjacent [PbBr₄] inorganic layers decrease from 10.09 Å (**1S**) to 10.04 Å (**1Rac**), consistent with the ones shown in the sing crystal structures (Fig. 2c, e). However, this result seems counterintuitive, that is, the larger the d value, the weaker the interaction between the cation and the inorganic anion and so the lower the $T_C$. To figure out this point, we further studied the single crystal structures and found that the N−H···Br distances in **1S** (2.50−2.72 Å) are shorter than the corresponding ones in **1Rac** (2.55 Å–2.91 Å) (Fig. 2c, e), meaning stronger N−H···Br interactions. This causes the lengthened distance of adjacent axial Br-defined planes from 3.78 Å (**1S**) to 3.92 Å (**1Rac**) and correspondingly the shortened distance of adjacent axial Pb-defined planes from 10.09 Å to 10.04 Å (Fig. 2c, e). Therefore, the [PbBr₆] octahedron in **1S** is more vertically distorted than **1Rac** (163.1° < 170.5°) (Fig. 2c, e) and shows greater inter-octahedral horizontal compression than **1Rac** (170.7° < 175.2°) (Fig. 2d, f), which further reduces the space for the cation to move and increases the rotational barrier. From the above analysis, the molecular chirality can significantly affect the octahedral distortions and local molecular packings, and thus the $T_C$ of the crystals.

## Role of enantiomer fraction engineering

The mechanochemical method, also known as solid-phase synthesis (SPS), can better control the stoichiometric ratio of the mixed components than the solution method[54,55]. To verify the effectiveness of this method, **1Rac** was prepared by using equal molar amounts of **1S**

and **1R** crystals. PXRD data show that the sample obtained by this method shows completely coincident peaks with the one obtained by the solution method using the racemic cation as a reactant (Supplementary Fig. 9a–c). So are the DSC results (Supplementary Fig. 9d). A series of $(S\text{-}3AMP)_x(R\text{-}3AMP)_{1-x}PbBr_4$ ($x = 0$–1 with an interval of every 0.05) was then synthesized by the SPS which ensures that the $x$-value is exactly the mixing ratio set during the synthesis (note: the enantiomeric fraction could not be determined experimentally for the prepared samples and the value is the expected enantiomer fraction). PXRD shows that there are both **1Rac** and **1S** structural features when $x$ is 0.65 and 0.70, which may be an intermediate phase or a mixture of the two phases (Fig. 3a and Supplementary Figs. 10 and 11a, b). To exclude the possibility of being a physical mixture, DSC and dielectric constant measurements were performed. As shown in Supplementary Fig. 11c, d, the curves of the physical mixture indicate the superimposed phase transition characteristics of both **1Rac** and **1S**. In contrast, the samples with $x$ 0.65 and 0.70 obtained by the SPS show characteristics of a single phase, rather than a mixture (Fig. 3b and Supplementary Fig. 12d, e). So, we suppose that these two ratios may be intermediate phases.

From Fig. 3b and Supplementary Fig. 12, it can be seen that the $T_C$ varies with the change of enantiomer fraction. We select the DSC data in the heating process to draw Fig. 3c. When $x$ increases from 0.5 to 0.65, the $T_C$ decreases from 359 K to 355 K; and when $x$ increases from 0.65 to 1, the $T_C$ increases from 355 K to 432 K (a change of 77 K), there is a nonlinear relationship between $x$ and $T_C$ (Fig. 3c and Supplementary Table 4). At the same time, by following the Boltzmann equation, $\Delta S = R \ln(N)$, the enthalpy change $\Delta H$, entropy change $\Delta S$, and orientation number $N$ with $x$ also show similar trends to the $T_C$ (Supplementary Table 5 and Fig. 3d). When $x = 0.70$, $\Delta H$, $\Delta S$, and $N$ have the minimum values, which are 4.4 kJ/mol, 12.2 J/(mol·K), 4.34, respectively, which are

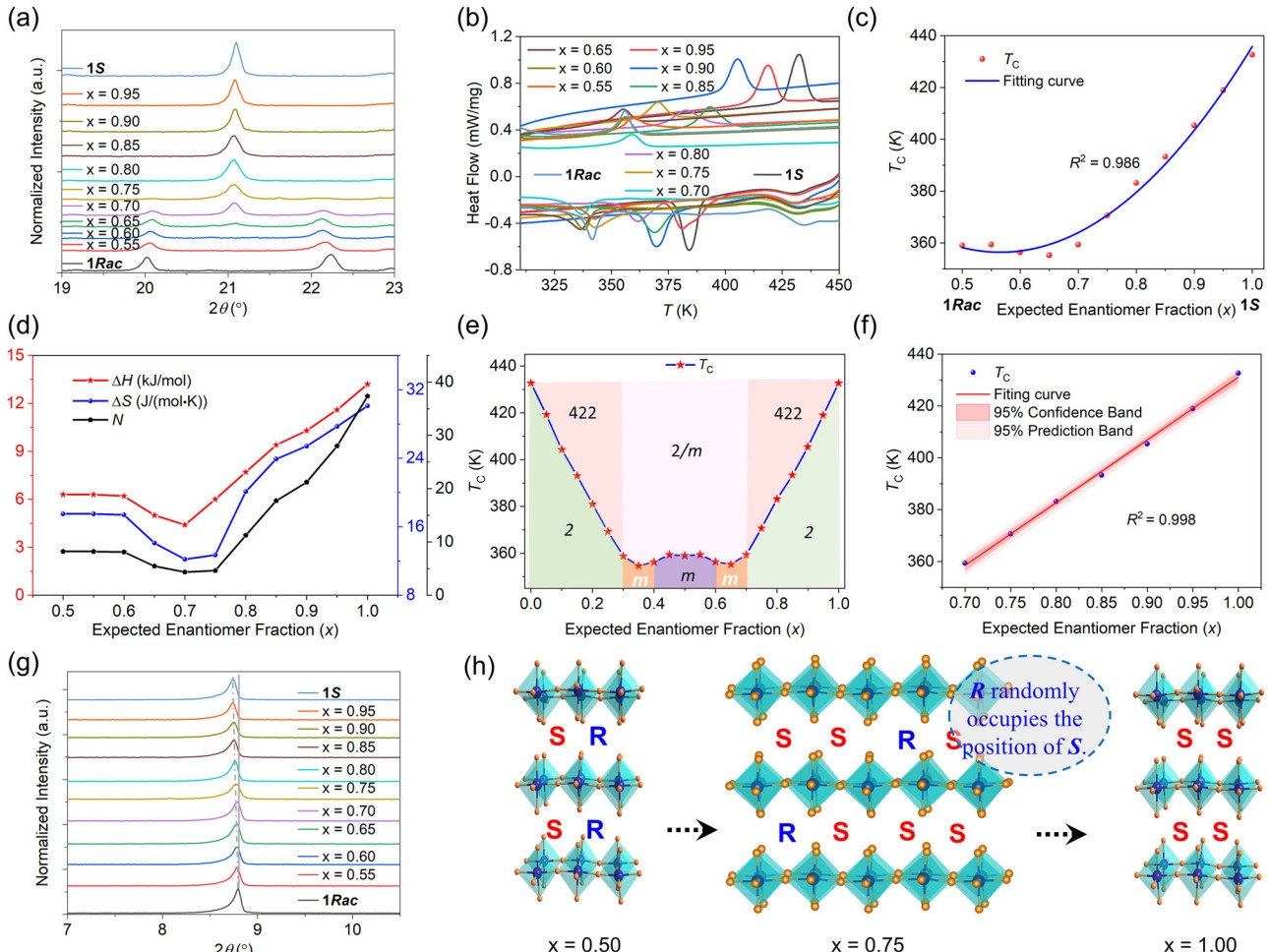

**Fig. 3 | Structures and phase transition properties of (S−3AMP)$_x$(R−3AMP)$_{1-x}$PbBr$_4$.** **a** PXRD patterns; **b** DSC curves; **c** Nonlinear correlation between the $T_C$ and $x$ (0.5–1). **d** Curves of enthalpy change, entropy change, and $N$ with $x$, $N$ is the proportion of the numbers of possible geometrical orientations allowed in the FEP and PEP. **e** Phase diagram and the point groups. The colors represent different crystallographic point groups, corresponding to the written point group symbols, and the orange region is roughly estimated due to the difficulty in obtaining ideal crystals. **f** Linear correlation between the $T_C$ and $x$ (0.70–1). The solid lines fit a linear regression model with the corresponding $R^2$ value, 95% confidence band, and 95% prediction band (shaded region). **g** PXRD peak representing the plane spacing of adjacent [PbBr$_4$] inorganic layers. **h** Schematic diagram of crystal structure change with enantiomer content.

characteristic of an order-disorder type of phase transitions. Therefore, we think that the enantiomer fraction does not change the phase transition type of the series. Furthermore, to verify the above experimental results, we synthesized samples with the $x$ between 0 and 0.5 (Supplementary Fig. 13). By extracting the DSC data in the heating process, it is found that the $T_C$ shows the same trend as $x = 0.5$–1 (Supplementary Fig. 13d).

Because the crystals obtained by SPS are micron crystalline powders and cannot be used for both single crystal structure and ferroelectric property analyses, we studied them through SHG (Supplementary Fig. 14a) and PXRD measurements (Supplementary Fig. 15). When $x ≤ 0.70$, the material undergoes a phase transition of 2/$mFm$ type while when $x > 0.70$, the phase transition is of 422$F$2 type (Fig. 3e), both of which comply with the Aizu rule. More importantly, when $x$ is in the range of 0.70–1, a strong linear correlation exists between the $x$ and $T_C$ (Fig. 3f). This ensures that the enantiomer fraction engineering strategy, as a simple and effective method, can continuously and linearly tune the composition and properties of chiral materials. The PXRD data show that when the $x$ increases from 0.5 to 1, the peak of the [PbBr$_4$] inorganic plane gradually moves from 8.80° to 8.76° (Fig. 3g), indicating that the $S$-cation can gradually replace the position of the $R$-

cation in **1Rac** (Fig. 3h). The increase in the enantiomer fraction affects the N−H···Br interactions in the crystal, which raises the rotation barrier for the organic cations to increase the $T_C$. The variation of $x$ also affects the SHG signals, showing a similar trend and a linear relationship with the change of $T_C$. These results demonstrate that enantiomeric fraction engineering is an effective means to tune the crystal packing structures and related properties.

As shown in Fig. 4a, we summarized the phase transition temperature data of some chiral and racemic systems. By comparison, we find that the temperature difference is a general phenomenon in both organic molecules and metal-halide systems[56–62]. It means that the nonequivalent intermolecular interactions of chiral molecules in crystals are universal. To further verify the effectiveness of the enantiomeric fraction engineering strategy, we selected (HTMPA)CdCl$_3$ as an example[63]. The synthesized (S-HTMPA)$_{0.75}$(R-HTMPA)$_{0.25}$CdCl$_3$ shows a phase transition temperature about 12 K higher than that of the enantiomeric pure one (S-HTMPA)CdCl$_3$ (Fig. 4b). Its phase purity is demonstrated by PXRD (Fig. 4c). Thus, we suppose that the enantiomeric fraction engineering strategy can be applied to chiral materials whose phase transition temperatures are different from that of their racemates.

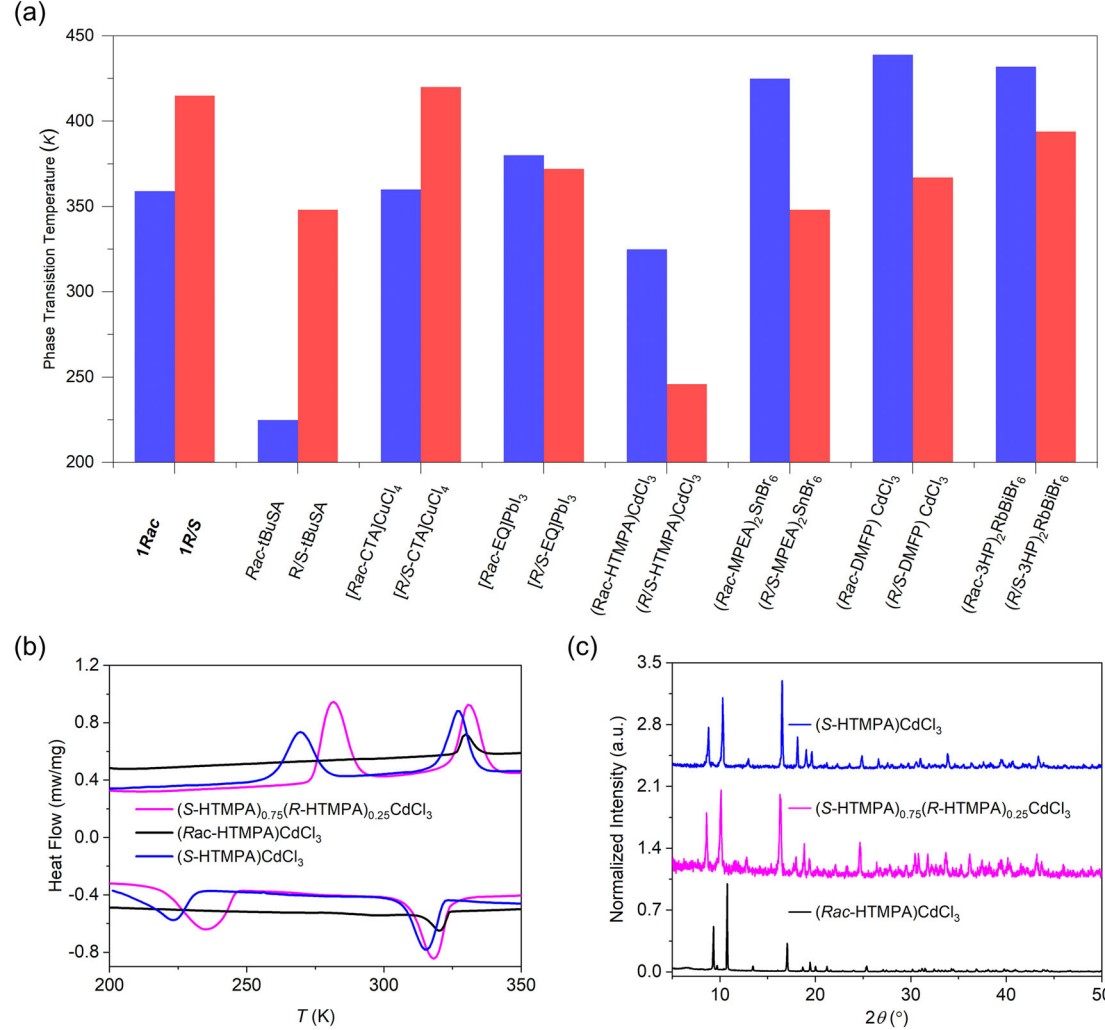

**Fig. 4 | Phase transition properties of chiral and racemic compounds.**
**a** Summary of phase transition temperatures of chiral and racemic compounds.
**b** DSC curves of $(S\text{-HTMPA})_x(R\text{-HTMPA})_{1-x}PbBr_4$ ($x = 0.5, 0.75$, and 1). **c** PXRD patterns of $(S\text{-HTMPA})_x(R\text{-HTMPA})_{1-x}PbBr_4$ ($x = 0.50, 0.75$, and 1.00). Note: tBuSA *tert*-butanesulfinamide[56], CTA 3-chloro-2-hydroxypropyltrimethylammonium[57], EQ *N*-ethylquinuclidinium[58], HTMPA hydroxy-*N,N,N*-trimethyl-2-propanaminium[63], MPEA *2*-phenyl-1-propylamineammonium[59], DMFP *N,N*-dimethyl-3-fluoropyrrolidinium[60], 3HP 3-hydroxypyrrolidinium[61,62].

## Semiconducting and chiroptical properties

Ultraviolet-visible absorption and diffuse reflectance spectra of **1Rac** are shown in Supplementary Fig. 16a. There is an obvious absorption in the ultraviolet region and the band edge extends to the visible region around 428 nm. The bandgap value (2.81 eV) is extracted from the low-energy slope of the Kubelka-Munk plot (Supplementary Fig. 16b). Density functional theory calculations show that **1Rac** is a direct bandgap semiconductor with a bandgap of 2.38 eV at the Γ point (Supplementary Fig. 16c). The partial density of states shows that the maximum value of the valence band is formed by the overlap of Pb-6s and Br-4p orbits, and the minimum value of the conduction band is formed by the overlap of Pb-6p orbits (Supplementary Fig. 16d). All these indicate that **1Rac** is a good optoelectronic semiconductor.

The non-centrosymmetric structure of lead-halide perovskites, combined with spin-orbit coupling, brings about a Rashba spin-splitting effect. With the variation of the enantiomer fraction $x$, $(S\text{-}3AMP)_x(R\text{-}3AMP)_{1-x}PbBr_4$ possesses the tunable bulk inversion asymmetry between chiral and nonchiral, making it a promising candidate for spintronics. Based on spin-dependent optical transition selection rules, CPL pumping can excite carriers with specific spin orientations, resulting in the difference of photoluminescence intensity ($P$) under the excitation of left- or right-handedness (L- or R-) of CPL[64,65]. The

degree of circular polarization of photoluminescence (PL) is evaluated by:

$$P = \left| \frac{I(\sigma^+) - I(\sigma^-)}{I(\sigma^+) + I(\sigma^-)} \right| \qquad (1)$$

where $I(\sigma^+)/I(\sigma^-)$ is the PL intensity with L/R-circular optical illumination.

We conducted CPL excited PL (CPLEPL) measurements on the serial samples, which all show large PL intensity differences (Fig. 5a and Supplementary Fig. 17). Crystalline-powdered sample of **1Rac** shows a CPLEPL difference of about 9.3% upon the L/R-CPL excitation at 395 nm and 298 K (Fig. 5b), which is comparable to the one of **1S** (Fig. 5c)[53]. This is consistent with the result that the asymmetry distortion of metal-halide octahedrons induced by the corresponding organic cations in the two structures causes the structural symmetry change[66]. To further rule out possible phase separation in $(S\text{-}3AMP)_x(R\text{-}3AMP)_{1-x}PbBr_4$ ($0.5 < x < 1$), we also measured the CPLEPL spectra of the physical mixture (molar ratio = 1:1) of **1Rac** and **1S**. It can be seen from Fig. 5d that the $P$-value of the mixture is 9.5%, just an intermediate between the values of **1Rac** and **1S**. In contrast, the $P$-value of $x = 0.75$ obtained by the SPS is 5.7% (Fig. 5e), about half of the one of the 1:1

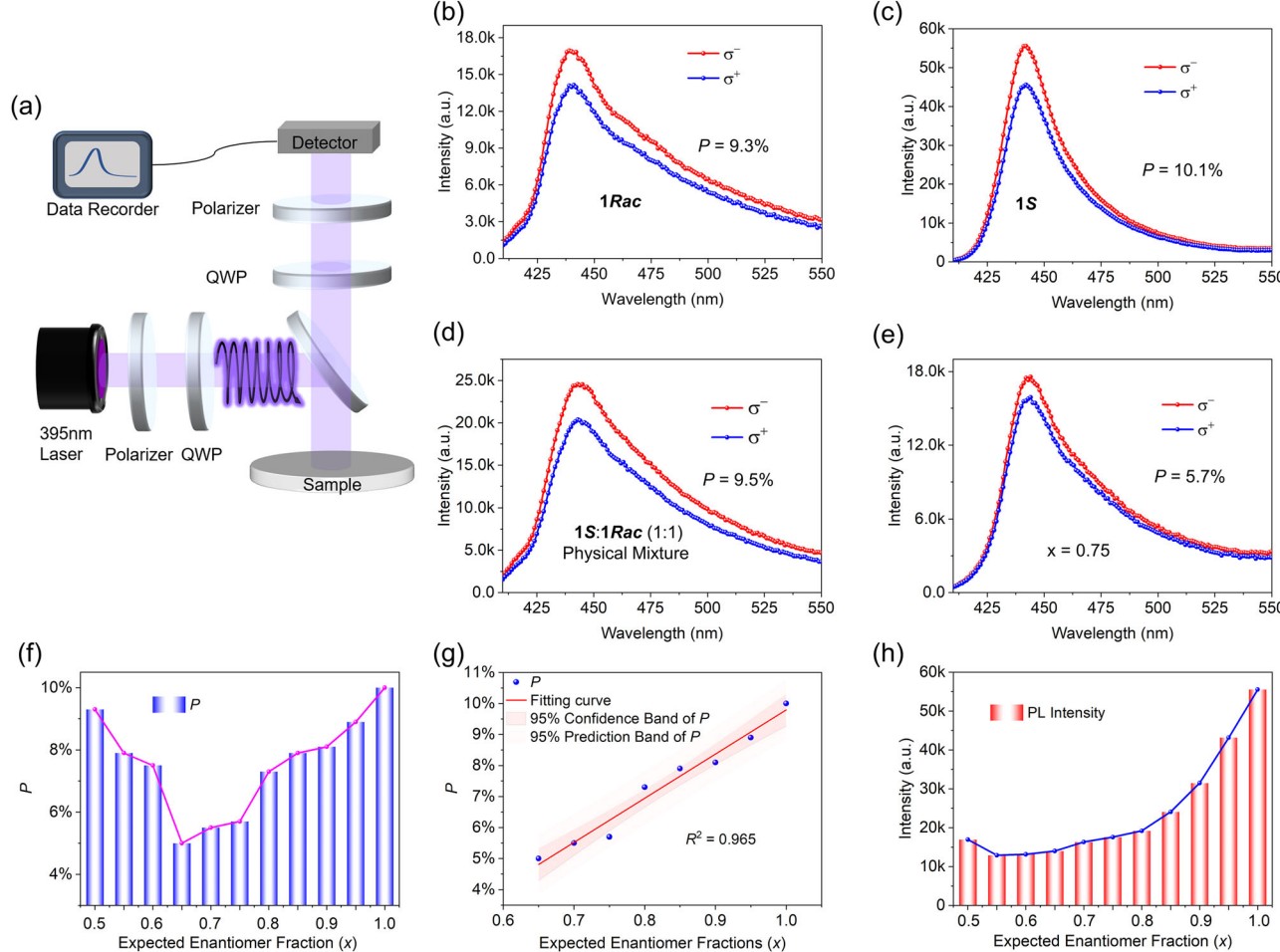

**Fig. 5 | CPLEPL spectra upon *L*-CPL (σ⁺) and *R*-CPL (σ⁻) excitation at 395 nm and 298 K. a** Schematic setup of the circularly polarization-sensitive PL spectroscopy. **b, c** CPLEPL spectra of **1*Rac*** ($x = 0.5$) and **1*S*** ($x = 1$). **d** CPLEPL spectra of the physical mixture (1:1) of **1*Rac*** and **1*S*; **e** CPLEPL spectra of the sample with $x = 0.75$; **f** Relationship between the $x$ and $P$. **g** Linear correlation between the $x$ and $P$. The solid lines fit a linear regression model with the corresponding $R^2$ value, 95% confidence band, and 95% prediction band (shaded region). **h** Relationship between the $x$ and PL intensity.

mixture. The $P$-values with $x$ between 0.5 and 1 are also significantly smaller than that of the mixture (Supplementary Fig. 17). By analyzing the $P$-values of all samples, we find that the relationship between the $x$ and $P$ is similar to the trend of the SHG intensity, that is, the $P$-value decreases when $x$ increases from 0.5 to 0.65 and then increases when $x$ from 0.65 to 1 (Fig. 5f and Supplementary Figs. 17 and 14c). There is a good linear correlation between $x$ (0.65–1) and $P$ (Fig. 5g). Both the $P$-value and the SHG strength are properties that are related to the non-centrosymmetric properties of the materials[27,67,68]. So we speculate that their microscopic origins can be ascribed to the changes of the non-centrosymmetry by enantiomer fraction-induced distortions of the inorganic sublattice (Fig. 3g and Supplementary Fig. 14c). Meanwhile, the enantiomer fraction also affects the PL intensity but not fluorescence lifetime, and the relationship between $x$ and PL intensity is similar to the trend of $T_C$ (Fig. 5h and Supplementary Figs. 18 and 19).

In conclusion, we have synthesized a series of hybrid metal-halide ferroelectrics by using an enantiomer fraction engineering strategy. With the change of the enantiomer fraction of the chiral cations, the phase transition temperature and related properties are smoothly tuned. A clear relationship between the enantiomer fraction and the respective property is established. This simple and effective strategy provides a way to expand chiral materials and regulate their structures and related properties.

## Methods

### Materials

All reagents were commercially available and used as received without further purification. *R*- and *S*-3AMP were purchased from WuXi AppTec with HPLC purity and chiral purity exceeding 99%. Stoichiometric amounts of (*R/S/Rac*)-3AMP and Pb(AcO)₂ were carefully dissolved in 50% HBr solution. A clear solution was obtained after continuous stirring for two hours at 373 K. Light yellow crystal powders were obtained after evaporation of the solution at 353 K for days. All the (*S*-3AMP)$x$(*R*-3AMP)$_{1-x}$PbBr₄ ($0 < x < 1$) perovskite powders were synthesized by grinding (*R*-3AMP)PbBr₄ (**1*R***) and (*S*-3AMP)PbBr₄ (**1*S***) crystals in a planetary ball mill containing four 50 mL reaction vessels for 24 h at 400 r/min. These samples can also be prepared from the racemic **1*Rac*** and enantiomer **1*R/S***.

### Single crystal and PXRD measurements

Crystallographic data were collected on a Rigaku Oxford Diffraction Supernova Dual Source, Cu at Zero equipped with an AtlasS2 CCD using Mo Kα radiation and an XtaLAB Synergy R, DW system, HyPix diffractometer. Rigaku CrysAlisPro software was used to collect data, refine cells, and reduce data. SHELXL-2018 with the OLEX2 interface was used to solve the structures by direct methods. All non-hydrogen atoms were refined anisotropically. The positions of hydrogen atoms

were generated geometrically. PXRD patterns were recorded on a Rigaku Ultima IV X-ray diffraction instrument with Cu K$\alpha$ radiation ($\lambda$ = 1.54056 Å). Diffraction patterns were collected in the 2$\theta$ range of 5–50° with a step size of 0.02°.

## Thermal analyses

Differential scanning calorimetry curves were recorded on a NETZSCH DSC 200F3 instrument at a scan rate of 20 K min$^{-1}$ under a nitrogen atmosphere, each sample was measured at least twice in succession. The thermogravimetric analysis curve was recorded on a NETZSCH TG209 F3 instrument with a heating rate of 20 K min$^{-1}$ under air atmosphere.

## Dielectric spectra

Samples are prepared by coating the conductive silver adhesive on surfaces as upper and lower electrodes. Temperature-dependent dielectric constant spectra were recorded by a Tonghui TH2828A impedance analyzer under the conditions of 1 ~ 1000 kHz frequency and 1 V external electric field.

## CPLEPL measurement

For circular-polarized pump and circular-polarized PL measurements, a quarter-waveplate and polarizer were placed in the path of the excitation beam, and another pair of polarization-dependent quarter-waveplate and polarizer was placed at the PL detection path. To exclude any instrumental polarization-dependent response, we kept the same detection polarization and only varied the incident polarization.

## Density functional theory (DFT) calculation

Density functional theory calculations were conducted by using the Vienna ab initio simulation package. Projector augmented wave method was used to define the ion-electron interactions. Exchange–correlation interaction was expressed by Perdew–Burke–Ernzerhof functional within the generalized gradient approximation. Grimme's dispersion-corrected semi-empirical DFT-D3 method was adopted to evaluate van der Waals interactions[69]. The energy cutoff was set to 500 eV and a 6 × 6 × 2 Monkhorst-Pack grid of k-points was used. VASPKIT was used to perform post-processing analysis[70].

## Reporting summary

Further information on research design is available in the Nature Portfolio Reporting Summary linked to this article.

# Data availability

All data generated and analyzed in this study are included in the article and the Supplementary Information, and are also available from corresponding authors upon request. The crystal structures generated in this study have been deposited in the Cambridge Crystallographic Data Center under accession code CCDC: 2257082 and 2257083 can be obtained free of charge from the CCDC via https://www.ccdc.cam.ac.uk/structures/.

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

## Acknowledgements

This work was financially supported by the National Natural Science Foundation of China (grants 21991144 and 22305034) and the Natural

Science Foundation of Jiangsu Province (BK20230811). We thank the Big Data Computing Center of Southeast University for providing the facility support on the calculations.

## Author contributions

C.-C.F. synthesized the samples, carried out the ferroelectric experiments and SHG experiments, and wrote the original manuscript. C.-D.L. solved and refined the crystal structures, and carried out the dielectric. B.-D.L., W.W., and M.-L.J. carried out the DSC experiments. C.-Q.J. and T.-Y.J. carried out the PXRD experiments. X.-B.H. built the CPLEPL test platform and carried out the CPLEPL characterizations. C.-Y.C. contributed to data analysis. W.Z. supervised the study, carried out the theoretical calculations, and revised the manuscript. All authors discussed the results and commented on the manuscript.

## Competing interests

The authors declare no competing interests.
