## [Peer Review File · Nature Communications]

REVIEWER COMMENTS

Reviewer #1 (Remarks to the Author):

The authors reported the successful tuning of phase transition temperature for ferroelectric crystals, $(S-3AMP)_x(R-3AMP)_{1-x}PbBr_4$ ($x = 0.61, 0.75, 0.81, \text{ and } 0.87$), by regulating the enantiomeric compositions of the chiral cations. They also found that the degree of circular polarization of PL increased with the increasing enantiomer fraction. These authors provide a simple method for the tuning of phase transition temperature and this is very interesting. However, I think this article is not novel enough because the chiral materials in this article have been published by them on *Advanced Materials* in 2022 and their CPL behaviors have been fully studied (*Adv. Mater.* 2022, 34, 2204119). After the elaborate tuning of the phase transition temperature, the physical properties of the materials such as dielectric constant, SHG intensity, and ferroelectricity with different enantiomer fractions do not show obvious improvement. Moreover, the physical characterizations of this series of materials $(S-3AMP)_x(R-3AMP)_{1-x}PbBr_4$ ($x = 0.61, 0.75, 0.81, \text{ and } 0.87$) are incomplete because the authors have not provided the single crystal structures, ferroelectric evidence, luminescence efficiency, etc. Nevertheless, I still think this work present an easy and novel method for the regulation of transition temperature and may provide new insight for other researchers. Thus, I look forward a revised version after substantial modification with a more thorough analysis of these materials and a deeper investigation of the regulation of phase transition temperature through the tuning of enantiomer fraction.

1. The results of tuning the phase transition through the regulation of enantiomeric excess are satisfactory. However, the characterization of the physical properties of $(S-3AMP)_x(R-3AMP)_{1-x}PbBr_4$ ($x = 0.61, 0.75, 0.81, \text{ and } 0.87$) is incomplete. A few additional points need to be added:

- How do the authors confirm the fraction of chiral components? If the enantiomer fraction is confirmed through the feeding ratio, then this result is not convincing. Direct proof through other methods is required.

- The single crystal structure and optical rotation of the material are suggested to be supplemented as it should provide more component information.

- Are these materials also ferroelectrics? If so, please provide the experimental proof.

- I think this series of materials is very similar to solid solutions. The phase diagram should be supplemented, including the high-temperature and low-temperature phases.

- Will there be a newly formed phase in the range of 0.6 to 0.75? The authors can refer to the reference (*Nat. Commun.* 2022, 13, 5329) for more details.

2. The P-E hysteresis loop of the racemic compound looks strange. Generally, there is only one current peak which corresponds to the polarization reversal. However, there is an additional current peak during the forward voltage application process. The author needs to provide a reasonable explanation. Meanwhile, I doubt the reliability of the ferroelectric behavior of these materials. The authors should provide other evidence for ferroelectricity such as PFM measurement.

3. The author analyzed the single crystal structure and found that 1-S has stronger hydrogen bonding and smaller molecular motion space compared to 1-Rac, resulting in a higher phase transition temperature. I do agree with the analyses and conclusions of the authors.

- However, the PXRD results contradict the above inference, and the author did not provide any reasonable explanation. Therefore, the authors are suggested to delete the corresponding description or provide a reasonable explanation for this contradiction.

- Besides, the authors should provide the crystal structures of this series of compounds and carefully compare the distance between adjacent inorganic layers to further support this inference.

4. The authors have investigated the CPL response of the chiral compounds in their previous work. In this work, the authors found that these compounds show circularly polarization-sensitive photoluminescence. Are there any changes in other parameters of the CPL performance of these materials with different molecular optical purity, such as quantum yield and anisotropy factor? Is there any difference in performance between mechanically mixed materials and those synthesized by the same proportion solution method?

5. In Figure 5h, why does the racemic compound show circularly polarized light-excited photoluminescence?

6. I doubt the usage of the term "engineering" by the authors. As mentioned by the author in the Introduction section, adjusting chiral purity is a neglected method. The enantiomer fraction engineering described by the author is more appropriate as a new method since its general validity

has not been confirmed. Besides, the difference between the phase transition temperatures of chiral crystals and racemic counterparts widely exists (Line 158). Does this method have a general validity? If so, it will be an interesting finding.

7. The logic of the second paragraph in the Introduction section is confusing.

8. In line 197, the authors claimed that "This is consistent with the result that the asymmetry distortion of metal halide octahedrons induced by the corresponding organic cations in the two structures causes the local reversal symmetry breaking (Table S4)." However, the authors have not provided Table S4 in the SI.

Minor points:

9. The "first" in line 64 should be deleted.

10. In Figure 2a, the peak of the blue line is partially missing.

11. Line 104, "(d)(e)" is a typo?

12. Figure S8(d)(e) is the result of which compound.

13. In Figure S10, the dielectric measurement of compound $x=0.75$ should be remeasured because its dielectric constant does not follow the change rule.

Reviewer #2 (Remarks to the Author):

For ferroelectric materials, the Curie temperature is one of the most essential parameters that determine the range of practical application. This manuscript submitted by Zhang describes a case study of the enantiomer fraction engineering as a promising means to tune the Curie temperatures of ferroelectric halide perovskite crystals, $(S-3AMP)_x(R-3AMP)_{1-x}PbBr_4$. When the enantiomer fraction x changes from 0.50 (racemate) to 1.0 (enantiomers), the continuous tuning of the Curie temperature was achieved. This tuning process gives rise to a tunable temperature range of about 60 K. From the design perspective of ferroelectrics, the subject and findings are certainly of importance for the related field, as well as breaking our conventional understanding on the role of compounds with low enantiomer fractions, that is, they are generally useless or even counterproductive. Therefore, these interesting results should be novel enough for the publication in Nature Communications in my opinion. However, the following issues need to be necessarily addressed.

(1). In addition to Curie temperature, ferroelectric materials have many other physical parameters, such as spontaneous polarization, residual polarization, coercive electric field, etc. For this series of halide perovskite crystals, $(S-3AMP)_x(R-3AMP)_{1-x}PbBr_4$, the phase transition temperature's linear tuning is well-established by changing the enantiomer fraction x . I wonder whether other ferroelectricity parameters also show the change with respect to the X values?

(2). As reported by authors, the enantiomer fraction x is very important for tuning the Curie temperatures. However, the differences between two enantiomers are quite small. How to determine the X value of the enantiomer fraction of the synthesized perovskite, and whether the X value is accurate or not? Is it feasible to change the crystal symmetry or crystal extinction?

(3). From the viewpoint of phase transition, the Curie temperature should closely relate to the energy barriers. Is this hypothetical also applied to this halide perovskite family? If so, the change of enantiomer fraction might also influence the phase transition energy barriers. Some discussions are recommended to add.

(4) Except for the ferroelectric properties, which physical property can be tuned by changing the enantiomer fraction, and which is not with this method?

(5) What about the phase stability of these halide perovskites in the air condition? Can they keep a long-term phase stability and anti-moisture? This is important for device practical application.

(6). Some formatting errors need to be checked and corrected, for example:

' $[PbBr_6]$ octahedron', ' $[PbBr_4]$ inorganic plane', and something like this.

In the second paragraph of "Semiconductor and chiroptical properties" section, "the non-centrosymmetric of the lead halide perovskite....." should be corrected as "the non-centrosymmetric structure of the lead halide perovskite".

The chemical formula should indicate its chemical valence. Please check the full text and revise it.

Reviewer #3 (Remarks to the Author):

In this manuscript, Fan et al. synthesized a series of two-dimensional Dion-Jacobson type lead bromide perovskite ferroelectrics $(S\text{-}3\text{AMP})_x(\text{R}\text{-}3\text{AMP})_{1-x}\text{PbBr}_4$ ($x = 0.5\text{--}1.0$) by changing the enantiomer fraction of chiral cations. They achieved continuous tuning of TC and established the relationship between chiral cation enantiomer fraction and TC. As we all know, TC is a key parameter in determining the high-temperature resistance of ferroelectric materials, and researchers have developed many methods to regulate TC. However, to my knowledge, the control of TC by enantiomeric fraction engineering of chiral cations has never been reported. In addition, the chiral optical properties can be regulated by enantiomeric fraction engineering. This is an exciting method, because few people in the field of physics and materials science research pay attention to it, or even ignore it. I believe that this work can arouse a wide range of interests and thoughts of researchers who focus on multiferroic and chiral optoelectronic materials. Therefore, I recommend that this manuscript be published by Nature Communications after minor revisions.

1. Does this methodology universally apply to all chiral and racemate compounds? Clarification on the broad scope of applicability across different compounds is essential.
2. Explain the observed non-linear trend in the change of P value depicted in Figure 5h. Identify underlying factors contributing to this deviation from linearity.
3. The ϵ' in the ordinate of Figure 2b and 2e should be revised as normalized ϵ' to distinguish it from Figure S10.

Reviewer #4 (Remarks to the Author):

The authors have presented an effective method for tuning the phase transition temperature using an enantiomer fraction engineering strategy. This approach assists in achieving the desired phase transition temperature range, enabling tailored applications at specific temperatures. It enhances the toolbox for ferroelectric-paraelectric tuning, serving as a valuable addition to the existing H/D and H/F strategies. These findings open the door to innovative advancements. However, certain aspects require careful consideration. By addressing these comments, the study's clarity, accuracy, and overall impact can be significantly enhanced, ensuring a comprehensive understanding of the enantiomer fraction engineering approach for tuning ferroelectric properties. The following comments emphasize areas in need of attention, without a specific order of significance. The paper could be accepted following a major revision.

1. The enantiomer doping method is different compared with H/D or H/F substitution method, chiral and racemate corresponding to H and F or H and D. Therefore, the introduction part should be revised. They are different objects and can't be compared. Your method corresponds to dope D to H contained compound from 0.5 to 1.
2. In Figure 3a, why the diffraction peak position is not linearly changed but a sudden change as X large than 0.61?
3. The phase transition temperature's linear tuning is well-established. Specifically, is there a linear shift in the Flack parameter, ranging from 0 for chiral compounds to 0.5 for racemate compounds?
4. Which physical property can be linearly tuned and which is not with this method?
5. It indicates that the spacing of adjacent $[\text{PbBr}_4]$ inorganic layers decreases from 10.09 Å of 1S to 10.04 Å of 1Rac (Figure 2b). In this sentence, Figure 2b does not contain the value of 10.09 and 10.04. Please check the figure citation.
6. What is the definition of angle beta in Figure 2? Are they the in-plane angle and out-of-plane angle? Please refer Chem. Sci., 2017, 8, 4497.
7. In Figure 2c-d, a lot of data is not mentioned in the main text. Please give more details about

them.

8. It is better to present the interesting results by drawing a phase transition temperature change diagram with R-rac-S sequence as X changes.

9. Can it get similar results by mechanically grinding the mixture of pure R and S compounds?

10. The grammar of the manuscript needs improved? Such as: Crystalline powdered samples of 1Rac show CPLEPL responses of about 8% for the L/R-CPL excitation at 395 nm and 298 K (Figure 5b), which is even comparable with those of R/S-(BPEA)₂PbI₄. It's better to revise them as follows: Crystalline powder sample of 1Rac shows a CPLEPL response difference of about 8% upon the L/R-CPL excitation xxxx.

11. Figure S7, S9, S11, and S12 are not mentioned in the main text. Please mark their position in the main text.

12. In addition to the dielectric, piezoelectric, pyroelectric, and nonlinear optical effects, the multiaxial feature stands out as another crucial property for assessing ferroelectricity. The compound mentioned in the report crystallizes in the Cc space group. However, the polarization direction is not specified in measuring the P-E loop. Please refer to a similar perovskite ferroelectric compound (Natl Sci Rev. 2020 Sep 8;8(5):nwaa232) with the sample space group.

Reviewer #5 (Remarks to the Author):

Key results

In the paper submitted by Fan et al., it is reported that the ferroelectric transition temperature of hybrid halide perovskite compounds could be tuned by adjusting the enantiomer fraction x of the chiral organic cations. Additionally, the experimental results demonstrate distinct circularly polarized light-excited photoluminescence (CPLEPL) responses (P) for compounds with different values of x.

Based on these experimental findings, the authors assert two main points:

- 1- A clear correlation was established between the enantiomer fraction x and the phase transition temperature T_c or P, with both exhibiting a linear relationship.
- 2- This novel "enantiomer fraction engineering" presents a facile and effective approach to investigating the structure and physical properties of chiral and polar perovskite compound assemblies.

Validity

Overview: The idea adopted as a strategy exhibits a certain degree of novelty and intrigue. While there are significant flaws in the experimental sample identification, the measurement results are clear. However, the analysis of the results is insufficient.

- 1- The experimental results concerning the crystal structures, ferroelectric properties, phase transition temperatures, and CPLEPL for both 1R/S (ref.50), previously reported by the authors, composed of enantiopure cations and 1Rac composed of racemic cations have been adequately described.
- 2- The most crucial experimental concern resides in the absence of a methodological exposition regarding the determination of the enantiomer fraction x of the obtained samples. Careful verification should be done to ascertain whether the fraction attained in the obtained samples aligns with the mixing ratios set during synthesis. It is important to elucidate how the value of x was determined. (The ensuing comments are all made under the assumption that the provided values of x are accurate.)
- 3- Regarding the claim "Line 57: the phase transitions, ~, showing a linear relationship between x and T_c or P." The experimental results indeed demonstrate different T_c and P values for each sample obtained through mixing. However, the proposed linear relationship by the authors between x and T_c, as presented in the form of T_c = 295 + 120x (Figure 3(h)), lacks clarity in

terms of the underlying analytical model and raises concerns about its validity. Furthermore, elucidation should be provided regarding the physical meanings of the values 295 and 120 obtained as parameters in this equation. Additionally, a linear relationship between x and P cannot be found from the presented experimental results.

Significance

1- There have been many reports discussing the examination of solids (= racemic solid solutions) obtained at various enantiomer fractions x . (For instance, *J. Am. Chem. Soc.* 2006, 128, 11985-11992, *Crystal Growth & Design* 2010, 10, 1808-1812.) Therefore, the novelty of the sample preparation method itself does not merit the proposition of a new nomenclature as a strategic innovation.

2- The investigation of the changes in physical properties (phase transition temperatures) resulting from incremental variations in x has primarily revolved around melting points until now. The application of this approach to the ferroelectric phase transition phenomenon and the experimental demonstration of actual changes in T_c offer novel contributions. Unfortunately, the authors have yet to present the material science significance or fundamental scientific insights into the ability to modulate phase transition temperatures "between racemic and R/S materials" using this approach. A more thorough analysis revealing similarities and discrepancies compared to the prior discussions on melting points would yield intriguing results.

3- As the authors suggest, the possibility of similar modulation in analogous compounds, as summarized in Figure 4, seems plausible. In the stage of implementing this in practical applications within society, the potential for such fine-tuning may be important.

Data and methodology

1- Figure 1 depicts several characteristics supporting a phase transition occurring at 359 K: (a) a rapid change in dielectric permittivity, (b) a peak in differential scanning calorimetry (DSC), (c) order-disorder transitions of cation molecules in the crystal structure, (d) a sudden rise in second harmonic generation (SHG), and (e) hysteresis in the polarization-electric field (P-E) curve. I see no major problems with these data.

2- In Figure 2, it becomes evident from the DSC peak at different temperatures (a) that 1S and 1Rac exhibit distinct phase transition temperatures. Although the peak areas appear significantly disparate, questions arise regarding whether the transition mechanisms could be regarded as identical for both. While the difference in rotational ease of the 3-AMP cation is attributed to the distortion of the PdBr6 octahedral structure and N-H-Br bond distances in single-crystal X-ray diffraction (SCXRD), offering a plausible explanation for the T_c disparity, it is also conceivable to discuss the same issue by comparing the volumes of the spaces occupied by the cations. Additionally, the variation in distortion of the PdBr6 octahedral structure presented here could potentially provide insights into the disparity in polarization values (0.50 uCcm⁻² for 1Rac and 1.0 uCcm⁻² for 1S/R).

3- In Figure 3, PXRD patterns (a) and (d) suggest crystallization in space group Cc for $x = 0.5$ and 0.61, while $x = 0.75$ -1.0 implies crystallization in space group P21. This indicates the formation of racemic solid solutions rather than mixtures of racemic and enantiopure crystals. The difference in phase transition temperatures can be confirmed from the DSC comparison (c). However, the lack of displayed vertical axis values prevents a comparison of peak intensities, thereby withholding essential information concerning the mechanism of the phase transition. Schematic diagram (g) for $x = 0.75$ gives the impression of highly ordered substitution to the R configuration within a crystal. In reality, wouldn't substitution within the solid solution occur randomly?

4- In Figure 5 (b)-(g), the alteration of CPLEPL spectra as x varies is notably clear.

5- Discrepancies arise between the experimental method outlined in the Supporting Information for single-crystal structure analysis and the contents of the submitted CIF file. Additionally, discrepancies also exist between the values in Table S1 and the content of the CIF file, particularly in parameters such as the Flack parameter and R1/wR2 values. These issues necessitate rectification.

Analytical approach

1- There seems to be room for discussion regarding the linear analysis in Figure 3(h). The PXRD results identify structural changes between Cc-C2/c for low fraction samples and between P21-P422 for high fraction samples. It does not seem appropriate to evaluate the phase transition temperatures (which are related to ΔH and ΔS) of such a system with different symmetry changes

using a simple linear correlation all together.

2- Figure 5(h) appears to lack any analysis or trends beyond the observed change in P with different enantiomer fractions x.

Suggested improvements

1- The actual determination of enantiomer fractions in the obtained racemic solid solutions is necessary. This could be achieved by redissolving the obtained solids, followed by extraction of the organic material and subsequent analysis using techniques such as chiral HPLC.

2- The peaks observed in DSC measurements should provide information about the entropy change (ΔH) associated with the phase transition. Given its significance when dealing with phase transition phenomena, a comparative analysis is crucial. Additionally, calculating the entropy (ΔS) might help assess whether the phase transition is truly order-disorder or displacement type.

3- An explanation is required for the observed changes in circularly polarized luminescence emission (CPLEPL) when P varies with different enantiomer fractions. Moreover, considering the definition of P, it seems to relate to enantiomeric excess (e.e.) rather than enantiomer fractions x. Plotting the x-axis of Figure 5(h) in terms of e.e. might aid in interpreting the results.

Clarity and context

1- I find it challenging to discern the clear purpose behind the authors' proposed methodology for tuning the ferroelectric phase transition temperature. Whether it aims to introduce a new fundamental scientific strategy, highlight novel phenomena, enhance material properties from a materials science perspective, or reduce environmental impact as part of green chemistry remains unclear. In the introduction, the authors criticize the approach of prior studies that observed changes in T_c , but the content of their critique lacks consistency. This has created ambiguity in the authors' stance towards this study.

2- The presentation of experimental results has generally been executed without significant issues. However, I find the descriptions regarding the interpretation and analysis of the results to be insufficient. Consequently, assertions such as the establishment of a "clear correlation between enantiomer fraction and phase transition temperature" and the "novelty and utility of the enantiomer fraction engineering approach" were difficult to interpret as having sufficient accuracy. I believe that a careful comparison with prior research results on the melting points of racemic solid solutions and a meticulous thermodynamic interpretation of DSC data would greatly enhance the quality of this study.

3- The results on the enantiomer fraction x dependency of CPLEPL are presented in conjunction. Nevertheless, their relevance to the title's assertion of "tuning the ferroelectric phase transition temperature" remains unclear.

4- Regarding terminology:

In L96, "the chiral & polar space group P21 (point group 2)" and L99, "chiral space group P422" are mentioned, yet P21 and P422 are not considered chiral space groups. Please verify the 22 unique chiral space groups (11 pairs) as listed in references such as J. Appl. Cryst. (2018). 51, 1481–1491.

The use of "homochiral" to convey enantiopure is not recommended by IUPAC and should be avoided. Exercise caution when employing "homochiral" concerning molecules. For reference, consult the IUPAC Gold Book entry on "enantiomerically pure (enantiopure)."

References

Unfortunately, the cited papers are somewhat inappropriate. Relying solely on examples from the domain of halide perovskites under the author's investigation when discussing fundamentally rooted and general phenomena such as phase transitions and well-established areas like the properties of ferroelectric materials is inadequate. Moreover, chirality has been well studied not only concerning enantioselective synthesis or pharmaceutical bioactivity but also more fundamental physical properties. The citation of references 48 and 49 in this context proves

difficult to comprehend. Notably absent is the consideration of prior research on phase transitions in chiral solid solutions.

My Expertise

I lack the expertise to evaluate the analysis and assessment of semiconductor properties through DFT calculations, as well as the interpretation of the computational methodology and its outcomes.

RESPONSE TO REVIEWERS' COMMENTS

Reviewer #1 (Remarks to the Author):

The authors reported the successful tuning of phase transition temperature for ferroelectric crystals, $(S-3AMP)_x(R-3AMP)_{1-x}PbBr_4$ ($x = 0.61, 0.75, 0.81, \text{ and } 0.87$), by regulating the enantiomeric compositions of the chiral cations. They also found that the degree of circular polarization of PL increased with the increasing enantiomer fraction. These authors provide a simple method for the tuning of phase transition temperature and this is very interesting. However, I think this article is not novel enough because the chiral materials in this article have been published by them on *Advanced Materials* in 2022 and their CPL behaviors have been fully studied (*Adv. Mater.* 2022, 34, 2204119). After the elaborate tuning of the phase transition temperature, the physical properties of the materials such as dielectric constant, SHG intensity, and ferroelectricity with different enantiomer fractions do not show obvious improvement. Moreover, the physical characterizations of this series of materials $(S-3AMP)_x(R-3AMP)_{1-x}PbBr_4$ ($x = 0.61, 0.75, 0.81, \text{ and } 0.87$) are incomplete because the authors have not provided the single crystal structures, ferroelectric evidence, luminescence efficiency, etc. Nevertheless, I still think this work presents an easy and novel method for the regulation of transition temperature and may provide new insight for other researchers. Thus, I look forward to a revised version after substantial modification with a more thorough analysis of these materials and a deeper investigation of the regulation of phase transition temperature through the tuning of enantiomer fraction.

Response:

Thanks for the comment!

1. The results of tuning the phase transition through the regulation of enantiomeric excess are satisfactory. However, the characterization of the physical properties of $(S-3AMP)_x(R-3AMP)_{1-x}PbBr_4$ ($x = 0.61, 0.75, 0.81, \text{ and } 0.87$) is incomplete. A few additional points need to be added:

Response:

We added more additional points with $x = 0 - 1$ (an interval of every 0.05).

1-1. How do the authors confirm the fraction of chiral components? If the enantiomer fraction is confirmed through the feeding ratio, then this result is not convincing. Direct proof through other methods is required. The single crystal structure and optical rotation of the material are suggested to be supplemented as it should provide more component information.

Response:

Thanks for these valuable suggestions!

Indeed, the proportion of products obtained by solution synthesis is not necessarily equal to the proportion of raw materials input. To address this issue, we considered methods such as post-analysis, that is, using techniques such as chiral high-performance liquid chromatography or optical rotation. However, the sample handling (dissolving, extraction, and purification) greatly affects the accuracy of the data. And the rotation angle of the enantiomers is very small and hard to detect.

Therefore, we adopted solid phase synthesis (SPS) to solve this problem successfully. This method can accurately control the stoichiometric ratio of the mixture and has become a main synthesis means to study the mixed composition perovskites (Science 2016, 354, 206; Chem. Commun. 2019, 55, 5079; iScience 2019, 16, 312–325; Chem. Mater. 2018, 30, 2309; Joule 2019, 3, 205; Adv. Energy Mater. 2020, 10, 1902499). For example, we synthesized **1Rac** by combining equal amounts of crystal powders of **1R** and **1S** in a planetary ball mill. The resulting sample shows the same structures and properties as **1Rac** crystal, proving the effectiveness of the solid-phase synthesis method (Supplementary Fig. 9).

Supplementary Fig. 9 | Properties of **1Rac synthesized by solution and solid phase synthesis.** (a) DSC comparison of **1Rac** synthesized by solution method and solid phase synthesis method. (b-d) PXRD patterns of **1Rac** synthesized by solution method and solid phase synthesis method.

Using this method, a series of samples with different enantiomer fractions were synthesized, and the fractional x value of the obtained samples was consistent with the mixture ratio set during synthesis. Subsequent experimental characterizations were performed on these newly synthesized samples. We then determined the relationship between the enantiomer fraction and phase transition temperature according to PXRD and DSC data (Fig. 3).

Fig. 3 | Structures and phase transition properties of $(S-3AMP)_x(R-3AMP)_{1-x}PbBr_4$. **a PXRD patterns; **b** DSC curves; **c** Nonlinear correlation between the T_C and x (0.5 – 1). **d** Curves of enthalpy change, entropy change, and N with x . **e** Phase diagram and the point groups. The orange region is roughly estimated, due to the difficulty in obtaining ideal crystals. **f** Linear correlation between the T_C and x (0.7– 1). The solid lines fit a linear regression model with the corresponding R^2 value, 95% confidence band, and 95% prediction band (shaded region). **g** PXRD peak representing the plane spacing of adjacent $[PbBr_4]$ inorganic layers. **h** Schematic diagram of crystal structure change with enantiomer content.**

1-2. Are these materials also ferroelectrics? If so, please provide the experimental proof.

Response:

They are ferroelectrics as shown in the main text:

Because the crystals obtained by SPS are micron crystalline powders and cannot be used for both single crystal structure and ferroelectric property analyses, we studied them through SHG (Supplementary Fig. 14a) and PXRD measurements

(Supplementary Fig. 15). When $x \leq 0.70$, the material undergoes a phase transition of $2/mFm$ type while when $x > 0.70$, the phase transition is of $422F2$ type (Fig. 3e), both of which comply with the Aizu rule.

Supplementary Fig. 14 | a SHG of the samples at 293 K.

Supplementary Fig. 15 | PXRD patterns of $(S-3AMP)_x(R-3AMP)_{1-x}PbBr_4$ ($x = 0.5 - 1$) at (a) 293 K and (b) 440 K.

Fig. 3e | Phase diagram and the point groups. The orange region is roughly estimated, due to the difficulty in obtaining ideal crystals.

1-3. I think this series of materials is very similar to solid solutions. The phase diagram should be supplemented, including the high-temperature and low-temperature phases.

Response:

Thanks for this valuable suggestion!

We have added a phase diagram (Fig. 3e).

1-4. Will there be a newly formed phase in the range of 0.6 to 0.75? The authors can refer to the reference (Nat. Commun. 2022, 13, 5329) for more details.

Response:

Thanks for this suggestion!

To determine whether a new phase is present between $x = 0.61$ and $x = 0.75$, we synthesized the samples ($x = 0.5 - 1$ with an interval of every 0.05) by solid-phase synthesis method as shown in the main text:

PXRD shows that there are **1*R*ac** and **1*S*** structural features when x is 0.65 and 0.70, which may be an intermediate phase or mixed phase (Fig. 3a, and Supplementary Fig. 10, 11a, 11b). To exclude the possibility of being a physical mixture, DSC and dielectric

constant measurements were performed. As shown in Supplementary Fig. 11c and 11d, the curves of the physical mixture indicate the superimposed phase transition characteristics of both **1*Rac*** and **1*S***. In contrast, the samples with $x = 0.65$ and 0.70 obtained by the SPS show characteristics of a single phase, rather than a mixture (Fig. 3b, and Supplementary Fig. 12d, 12e). So, we suppose that these two ratios may be intermediate phases.

To verify the above experimental results, we also synthesized samples with enantiomer fraction $x = 0 - 0.5$, and performed DSC and PXRD tests (Supplementary Fig. 13). They show the same trend as the series ($x = 0.5 - 1$). Based on the above results, we suppose the phase at $x = 0.65/0.35$ and $0.70/0.30$ is an intermediate phase.

Supplementary Fig. 10 | PXRD study of $(S-3AMP)_x(R-3AMP)_{1-x}PbBr_4$ ($x = 0.5 - 1$) at 293 K.

Supplementary Fig. 11 | Properties of physical mixture of 1Rac and 1S at 293 K. a PXRD pattern. **b** Enlarged PXRD pattern. **c** DSC curve. **d.** Real part of the dielectric constant.

Supplementary Fig. 12 | d, e Real part of the dielectric constant of $(S-3AMP)_x(R-3AMP)_{1-x}PbBr_4$ ($x = 0.65$ and 0.70).

Supplementary Fig. 13 | Structure and phase transition properties of $(S-3AMP)_x(R-3AMP)_{1-x}PbBr_4$ ($x = 0 - 0.5$). **a PXRD patterns. **b** DSC curves. **c** Nonlinear correlation between the phase transition temperature and x . **d**. Relationship between phase transition temperature and x ($0 - 1$) during the heating process of DSC.**

2. The P–E hysteresis loop of the racemic compound looks strange. Generally, there is only one current peak which corresponds to the polarization reversal. However, there is an additional current peak during the forward voltage application process. The author needs to provide a reasonable explanation. Meanwhile, I doubt the reliability of the ferroelectric behavior of these materials. The authors should provide other evidence for ferroelectricity such as PFM measurement.

Response:

The P–E hysteresis loop is not strange. Ferroelectricity is an insulator property, but our compounds are semiconductors. Their leakage current and defects can cause the imperfect loop (see James F. Scott. "Ferroelectrics go bananas." J. Phys.: Condens. Matter, 2008, **20**, 021001). Due to the poor solubility of our material and the adverse stress release during crystal growth, we have not been able to obtain high-quality crystals, which can also hurt testing. The flip of the polarization current observed in the J – E diagram is favorable evidence for the ferroelectric properties.

3. The author analyzed the single crystal structure and found that 1-S has stronger hydrogen bonding and smaller molecular motion space compared to 1-Rac, resulting in a higher phase transition temperature. I do agree with the analyses and conclusions of the authors.

Response:

Thanks!

3-1. However, the PXRD results contradict the above inference, and the author did not provide any reasonable explanation. Therefore, the authors are suggested to delete the corresponding description or provide a reasonable explanation for this contradiction.

Response:

Fig. 3g, 3h | PXRD peak representing the $[PbBr_4]$ inorganic plane of $(S-3AMP)_x(R-3AMP)_{1-x}PbBr_4$ ($x = 0.5 - 1$). **h** Schematic diagram of crystal structure change with enantiomer content.

We explained this in the main text:

The PXRD data show that when the x increases from 0.5 to 1, the peak of the $[PbBr_4]$ inorganic plane gradually moves from 8.80° to 8.76° (Fig. 3g), indicating that the S -cation can gradually replace the position of R -cation in **1Rac** (Fig. 3h). The increase in the enantiomer fraction affects the $N-H\cdots Br$ interactions in the crystal, which gives the organic cations a higher motion barrier to increase the T_C .

3-2. Besides, the authors should provide the crystal structures of this series of compounds and carefully compare the distance between adjacent inorganic layers

to further support this inference.

Response:

A good suggestion!

We tried this characterization several times but failed. These compounds synthesized by the solution method are all obtained at high temperatures. Because of their low solubility, we cannot get crystals with good quality for structural analysis. The micro-crystalline powders obtained by solid-phase synthesis cannot be used for single-crystal structural analysis. We expect to solve this problem in future work.

4-1. The authors have investigated the CPL response of the chiral compounds in their previous work. In this work, the authors found that these compounds show circularly polarization-sensitive photoluminescence. Are there any changes in other parameters of the CPL performance of these materials with different molecular optical purity, such as quantum yield and anisotropy factor?

Response:

We tested the quantum yield and fluorescence lifetime of the series. As shown below, the quantum yield values are all negative due to the low signal-to-noise ratio caused by the weak luminescence of the series, consistent with the fact that they emit very weak fluorescence. Consequently, the luminescent anisotropy factor has not been studied. As can be seen from Figure S17, the fluorescence lifetimes of the materials are all between 0.38 and 0.49 ns, and there is no significant correlation between the fluorescence lifetimes and the enantiomer fraction x .

PLQY of the synthesized materials at 298 K. (a-c) PLQY of $(S\text{-}3\text{AMP})_x(R\text{-}3\text{AMP})_{1-x}$ ($x = 0.5, 0.75, \text{ and } 1$) at 298 K.

Supplementary Fig. 19 PL decay curves of the series at 298 K. a–k PL decay curves of $x = 0.5 - 1$ at 298 K. l Comparison of the PL lifetime τ values with different enantiomer fractions.

4-2 Is there any difference in performance between mechanically mixed materials and those synthesized by the same proportion solution method?

Response:

Because the proportion of the sample made by the solution method is different from the solid-phase synthesized sample, they can not be directly compared.

For consistency, we adopted the solid-phase synthesized samples for structural and

property studies because they have the accurate stoichiometric ratio as the input reactants, but the solution method does not.

5. In Figure 5h, why does the racemic compound show circularly polarized light-excited photoluminescence?

Response:

The non-centrosymmetry in the polar ferroelectric crystals (not necessarily chiral), together with a strong spin–orbit coupling ensured by the heavy atoms, gives rise to spin-splitting bands (Rashba effect). Therefore, the excitation of circularly polarized light will result in fluorescence with different intensities. This phenomenon generally exists in non-centrosymmetric lead-halide perovskites, as is the case of our racemic compound **1*Rac***. (see *J. Am. Chem. Soc.* 2019, 141, 15972–15976; *Nano Lett.* 2021, 21, 4584–4591; *Nat. Commun.* 2019, 10, 484)

6-1. I doubt the usage of the term “engineering” by the authors. As mentioned by the author in the Introduction section, adjusting chiral purity is a neglected method. The enantiomer fraction engineering described by the author is more appropriate as a new method since its general validity has not been confirmed.

Response:

We summarized and tested examples to show that enantiomer fraction engineering is a new method of tuning structures and properties. It is surely neglected by materials researchers.

The related contents in the main text are shown below:

Fig. 4 | Phase transition properties of chiral and racemic materials. a Summary of phase transition temperatures of chiral and racemic materials. **b** DSC curves of (S-HTMPA)_x(R-HTMPA)_{1-x}PbBr₄ ($x = 0.5, 0.75,$ and 1). **c** PXRD of (S-HTMPA)_x(R-HTMPA)_{1-x}PbBr₄ ($x = 0.5, 0.75,$ and 1).

As shown in **Fig. 4a**, we summarized the phase transition temperature data of some chiral and racemic systems. By comparison, we find that the temperature difference is a general phenomenon in both organic molecules and metal-halide systems.⁵⁶⁻⁶² It means that nonequivalent hydrogen bond interactions of chiral molecules in crystals are universal. To further verify the effectiveness of the enantiomeric fraction engineering strategy, we selected (HTMPA)CdCl₃ as an example.⁶³ The synthesized (S-HTMPA)_{0.75}(R-HTMPA)_{0.25}CdCl₃ shows a phase transition temperature 12 K higher than that of the enantiomeric pure one (S-HTMPA)CdCl₃ (Fig. 4b). Its phase purity is

further demonstrated by PXRD (Fig. 4c). Thus, we suppose that the enantiomeric fraction engineering strategy can be applied to all chiral materials whose phase transition temperatures are different from that of their racemates.

6-2. Besides, the difference between the phase transition temperatures of chiral crystals and racemic counterparts widely exists (Line 158). Does this method have a general validity? If so, it will be an interesting finding.

Response:

We described this point in the main text:

As shown in Fig. 4a, we summarized the phase transition temperature data of some chiral and racemic systems. By comparison, we find that the temperature difference is a general phenomenon in both organic molecules and metal-halide systems.⁵⁶⁻⁶² It means that nonequivalent hydrogen bond interactions of chiral molecules in crystals are universal. To further verify the effectiveness of the enantiomeric fraction engineering strategy, we selected (HTMPA)CdCl₃ as an example.⁶³ The synthesized (*S*-HTMPA)_{0.75}(*R*-HTMPA)_{0.25}CdCl₃ shows a phase transition temperature 12 K higher than that of the enantiomeric pure one (*S*-HTMPA)CdCl₃ (Fig. 4b). Its phase purity is further demonstrated by PXRD (Fig. 4c). Thus, we suppose that the enantiomeric fraction engineering strategy can be applied to all chiral materials whose phase transition temperatures are different from that of their racemates.

Based on the current research and statistics, we believe that this strategy has universal applicability.

7. The logic of the second paragraph in the Introduction section is confusing.

Response:

We revised this part in the introduction section:

In molecular ferroelectric crystals, the shape and size of the molecules greatly regulate the packing structures, which makes it possible to optimize the ferroelectricity through rational molecular design.⁴³ However, due to the highly sensitive nature of ferroelectricity to molecular structure and intermolecular interactions, sometimes, even

substituting a single atom can lead to the disappearance of ferroelectricity. Chirality has the advantage of having nearly the same molecular interactions with counterions in the crystal structures, which can allow finely tuning of the T_C by maintaining the ferroelectricity. Moreover, as a basic molecular structure feature, chirality has been exploited to design and construct chiral&ferroelectric materials to achieve bulk photovoltaic effect,⁴⁴ chiroptical-coupled ferroelectric properties such as circularly polarized luminescence,^{45,46} and circularly polarized light (CPL) detection.⁴⁷ However, in these studies, little attention has been paid to the impact of enantiomeric excess on the structures and properties of the ferroelectrics, and the use of molecules with different chiral purities has never been considered when developing new materials. In the past, research on enantiomer solid solutions mostly focused on the difference in solubility and melting point of compounds with different enantiomer fractions of organic molecular crystals, which is generally used to guide chiral separation.^{48,49} In organic chemistry, how to get enantiomerically pure molecules is always the focus because compounds with different enantiomeric purities often show different biological activities.^{50,51} Generally, the potentials of materials with different enantiomer fractions have been overlooked in the discovery of new structures and properties as well as the cost consideration in material processing.

8. In line 197, the authors claimed that “This is consistent with the result that the asymmetry distortion of metal halide octahedrons induced by the corresponding organic cations in the two structures causes the local reversal symmetry breaking (Table S4).” However, the authors have not provided Table S4 in the SI.

Response:

Thanks!

We have corrected the error.

Minor points:

9. The “first” in line 64 should be deleted.

Response

We deleted the “first” and the sentence was revised as:

Single crystals of the enantiomer racemate **1Rac** and **1S** were synthesized by a solution-evaporation method.

10. In Figure 2a, the peak of the blue line is partially missing.

Response:

We remeasured the DSC of **1S** and **1Rac**, and the revised picture is as follows:

11. Line 104, “(d)(e)” is a typo?

We revised the sentence as:

d and **e** N–H···Br interactions, out-of-plane, and in-plane distortions of **1Rac** at 293 K.

12. Figure S8(d)(e) is the result of which compound.

Response:

Figure S8(d)(e) is the result of the **1S** compound that was published in *Advanced Materials* (*Adv. Mater.* 2022, 34, 2204119).

The sentence was revised as:

(d) dielectric transition of **1S**; **(e)** SHG signal of **1S**.

13. In Figure S10, the dielectric measurement of compound $x=0.75$ should be remeasured because its dielectric constant does not follow the change rule.

Response:

The dielectric measurement of compound $x=0.75$ was remeasured and the picture was renumbered as Supplementary Fig 12f.

Reviewer #2 (Remarks to the Author):

For ferroelectric materials, the Curie temperature is one of the most essential parameters that determine the range of practical applications. This manuscript submitted by Zhang describes a case study of the enantiomer fraction engineering as a promising means to tune the Curie temperatures of ferroelectric halide perovskite crystals, $(S-3AMP)_x(R-3AMP)_{1-x}PbBr_4$. When the enantiomer fraction x changes from 0.50 (racemate) to 1.0 (enantiomers), the continuous tuning of the Curie temperature was achieved. This tuning process gives rise to a tunable temperature range of about 60 K. From the design perspective of ferroelectrics, the subject and findings are certainly of importance for the related field, as well as breaking our conventional understanding on the role of compounds with low enantiomer fractions, that is, they are generally useless or even counterproductive. Therefore, these interesting results should be novel enough for the publication in Nature Communications in my opinion. However, the following issues need to be necessarily addressed.

Response:

Thanks!

(1). In addition to Curie temperature, ferroelectric materials have many other physical parameters, such as spontaneous polarization, residual polarization, coercive electric field, etc. For this series of halide perovskite crystals, $(S-3AMP)_x(R-3AMP)_{1-x}PbBr_4$, the phase transition temperature's linear tuning is well-established by changing the enantiomer fraction x . I wonder whether other ferroelectricity parameters also show the change with respect to the X values?

Response:

The series synthesized by solution method is obtained at high temperature. Because of their low solubility, we cannot get enough quality crystals for structure studies and so are the crystal-based ferroelectric properties. We expect to solve this problem in future work.

We have shown that, besides the phase transition temperature, some other properties such as second-harmonic generation intensity, degree of circular polarization of photoluminescence, and photoluminescence intensity of the materials have also been tuned.

(2). As reported by authors, the enantiomer fraction x is very important for tuning the Curie temperatures. However, the differences between two enantiomers are quite small. How to determine the X value of the enantiomer fraction of the synthesized perovskite, and whether the X value is accurate or not? Is it feasible to change the crystal symmetry or crystal extinction?

Response:

Thanks for these valuable comments!

Since the proportion of products obtained by solution synthesis is not necessarily equal to the proportion of raw materials input. For this reason, we also consider using other methods to solve this problem, such as trying to redissolve the obtained solids, then extract the organic matter and subsequently confirm the analysis using techniques such as chiral high-performance liquid chromatography, or by directly measuring the optical rotation of perovskite solution. However, two-step synthesis, extraction, and purification will greatly affect the accuracy of the data. At the same time, because the Angle of rotation of the enantiomer pure molecule is very small, it will affect the determination of perovskite with a low enantiomer fraction.

The mechanochemical method, also known as solid-phase synthesis (SPS), can better control the stoichiometric ratio of the mixed components than the solution method (Adv. Energy Mater. 2020, 10, 1902499), and become the main synthesis method to study the basic properties of the mixed composition perovskite (Science 2016, 354, 206; Chem. Commun. 2019, 55, 5079; iScience 2019, 16, 312–325; Chem. Mater. 2018, 30, 2309; Joule 2019, 3, 205).

We put 0.5g **1R** crystal powders and 0.5g **1S** crystal powders into a planetary ball mill containing four 50ml reaction vessels (MITR yxqm-0.4l stainless steel grinding tank in a roll of 50ml and 1 stainless steel ball diameter of 10mm, 2 stainless steel ball diameter

of 8mm, and 27 stainless steel ball diameter of 5mm), Running at 400 r/min for 24 hours yields **1Rac**. DSC shows that 1Rac obtained by both methods has the same phase transition temperature, and PXRD further confirms its phase purity. The experiment proves that we can obtain the target compound by solid phase synthesis method (Supplementary Fig. 9).

Supplementary Fig. 9 | Properties of 1Rac synthesized by solution and solid phase synthesis. a-c PXRD patterns of 1Rac synthesized by solution method and solid-phase synthesis method. d DSC comparison of 1Rac synthesized by solution method and solid-phase synthesis method.

A series of samples with different enantiomer ratios was synthesized by the solid-phase synthesis method, and the fractional x value of the obtained samples was consistent with the mixture ratio set during synthesis. Subsequent experimental characterization was performed based on the newly synthesized sample.

(3). From the viewpoint of phase transition, the Curie temperature should closely relate to the energy barriers. Is this hypothetical also applied to this halide perovskite family? If so, the change of enantiomer fraction might also influence the phase transition energy barriers. Some discussions are recommended to add.

Response:

This is a valuable suggestion!

But there are some difficulties to hinder us from doing so. We do not have the well-solved single-crystal structure of the series. The theoretical calculations are not efficient for differentiating the tiny structural variation, especially the packing structures dominated by delicately balanced intermolecular interactions.

(4) Except for the ferroelectric properties, which physical property can be tuned by changing the enantiomer fraction, and which is not with this method?

Response:

According to the present experimental results, we have shown that the phase transition temperature, enthalpy change, dielectric, SHG intensity, PL intensity, and chiral properties of the material can be regulated by this method. We also suppose the parameters such as coercive field, residual polarization value, and saturation polarization value can be further tuned. However, due to the lack of single crystals, we have not been able to measure them presently.

(5) What about the phase stability of these halide perovskites in the air condition? Can they keep a long-term phase stability and anti-moisture? This is important for the device's practical application.

Response:

These halide perovskites have good phase stability under air conditions. The thermal stability was tested under an air atmosphere, showing that the powder sample has good air and thermal stability below 610 K.

(6). Some formatting errors need to be checked and corrected, for example:

'[PbBr₆] octahedron', '[PbBr₄] inorganic plane', and something like this.

In the second paragraph of "Semiconductor and chiroptical properties" section, "the non-centrosymmetric of the lead halide perovskite...." should be corrected as "the non-centrosymmetric structure of the lead halide perovskite".

The chemical formula should indicate its chemical valence. Please check the full text and revise it.

Response:

Thank!

We checked the full text and revised these errors.

Reviewer #3 (Remarks to the Author):

In this manuscript, Fan et al. synthesized a series of two-dimensional Dion-Jacobson type lead bromide perovskite ferroelectrics $(S\text{-}3\text{AMP})_x(\text{R}\text{-}3\text{AMP})_{1-x}\text{PbBr}_4$ ($x = 0.5\text{--}1.0$) by changing the enantiomer fraction of chiral cations. They achieved continuous tuning of TC and established the relationship between chiral cation enantiomer fraction and TC. As we all know, TC is a key parameter in determining the high-temperature resistance of ferroelectric materials, and researchers have developed many methods to regulate TC. However, to my knowledge, the control of TC by enantiomeric fraction engineering of chiral cations has never been reported. In addition, the chiral optical properties can be regulated by enantiomeric fraction engineering. This is an exciting method, because few people in the field of physics and materials science research pay attention to it, or even ignore it. I believe that this work can arouse a wide range of interests and thoughts of researchers who focus on multiferroic and chiral optoelectronic materials. Therefore, I recommend that this manuscript be published by Nature Communications after minor revisions.

1. Does this methodology universally apply to all chiral and racemate compounds?
Clarification on the broad scope of applicability across different compounds is essential.

Response:

As shown in Fig. 4a, we summarized the phase transition temperature data of some chiral and racemic systems of different categories according to the literature. By comparison, we find that there are examples of phase transition temperature differences between chiral and racemic crystals in both organic molecular or metal halide systems. This also indicates that asymmetric hydrogen bond interactions of chiral molecules in crystals are universal. Enantiomeric solid solutions in organic molecular crystals have also been reported experimentally, which proves the existence of the low enantiomeric component phase of the system and the feasibility of the method. To further verify the

generality of the strategy, we selected (HTMPA)CdCl₃ as an example to verify the metal halide system. The compound (S-HTMPA)_{0.75}(R-HTMPA)_{0.25}CdCl₃ was synthesized, and its phase transition temperature was 12 K higher than that of the enantiomeric pure compound (S-HTMPA)CdCl₃ (Fig. 4b). PXRD demonstrated its phase purity (Fig. 4c), which further demonstrated the effectiveness of this strategy experimentally. Based on the present study, we believe that this strategy is generally suitable for chiral and racemic compounds with different phase transition temperatures. However, the temperature increase and decrease depend on the phase transition temperature of chiral crystals and racemic crystals.

Fig. 4 | Phase transition properties of chiral and racemic materials.

2. Explain the observed non-linear trend in the change of P value depicted in Figure 5h. Identify underlying factors contributing to this deviation from linearity.

Response:

We explained this in the main text:

By analyzing the P values of all samples, we find that the relationship between the x and P is similar to the trend of the SHG intensity, that is, the P value decreases when x increases from 0.5 to 0.65 and then increases when x from 0.65 to 1.0 (Fig. 5f and Supplementary Fig. 17 and 14c). There is a good linear correlation between x (0.65 – 1) and P (Fig. 5g). Both the P value and the SHG strength are properties that are related to the non-centrosymmetric properties of the materials.^{27,67,68} So we speculate that their microscopic origins can be ascribed to the changes of the non-centrosymmetry by enantiomer fraction-induced distortions of the inorganic sublattice (Fig. 3g and Supplementary Fig. 14c).

References

27. Park, I. H.; Zhang, Q. N.; Kwon, K. C.; Zhu, Z. Y.; Yu, W.; Leng, K.; Giovanni, D.; Choi, H. S.; Abdelwahab, I.; Xu, Q. H.; Sum, T. C.; Loh, K. P. Ferroelectricity and rashba effect in a two-dimensional dion-jacobson hybrid organic-inorganic perovskite. *J. Am. Chem. Soc.* 141, 15972-15976 (2019).
67. Noma, T.; Chen, H. Y.; Dhara, B.; Sotome, M.; Nomoto, T.; Arita, R.; Nakamura, M.; Miyajima, D. Bulk photovoltaic effect along the nonpolar axis in organic-inorganic hybrid perovskites. *Angew. Chem. Int. Ed.* 62, e202309055 (2023).
68. B. I. Sturman; V. M. Fridkin, *The photovoltaic and photorefractive effects in noncentrosymmetric materials.* (Routledge, London,2021)

Supplementary Fig. 17. | Studies of Circularly polarization-sensitive PL spectroscopy at 298 K.

Supplementary Fig. 14c | Comparison of the P values of the perovskites with different enantiomer fractions.

Fig. 5g | Linear correlation between the P and x . The solid lines fit a linear regression model with the corresponding R^2 value, 95% confidence band, and 95% prediction band (shaded region).

Fig. 5g | PXRD peak representing the $[\text{PbBr}_4]^{2-}$ inorganic plane of $(S\text{-}3\text{AMP})_x(R\text{-}3\text{AMP})_{1-x}\text{PbBr}_4$ ($x = 0.00 - 0.50$)

3. The ϵ' in the ordinate of Figure 3b and 3e should be revised as normalized ϵ' to distinguish it from Figure S10.

Response:

Thanks!

We revised it.

Reviewer #4 (Remarks to the Author):

The authors have presented an effective method for tuning the phase transition temperature using an enantiomer fraction engineering strategy. This approach assists in achieving the desired phase transition temperature range, enabling tailored applications at specific temperatures. It enhances the toolbox for ferroelectric-paraelectric tuning, serving as a valuable addition to the existing H/D and H/F strategies. These findings open the door to innovative advancements. However, certain aspects require careful consideration. By addressing these comments, the study's clarity, accuracy, and overall impact can be significantly enhanced, ensuring a comprehensive understanding of the enantiomer fraction engineering approach for tuning ferroelectric properties. The following comments emphasize areas in need of attention, without a specific order of significance. The paper could be accepted following a major revision.

1. The enantiomer doping method is different compared with H/D or H/F substitution method, chiral and racemate corresponding to H and F or H and D. Therefore, the introduction part should be revised. They are different objects and can't be compared. Your method corresponds to dope D to H contained compound from 0.5 to 1.

Response:

We revised this part in the introduction section:

Furthermore, H/D isotope and H/F substitution are either 0 or 1 doping, while two components linear doping in tuning the ferroelectricity and related properties, are rarely explored, some new phenomena may be missed.

2. In Figure 3a, why the diffraction peak position is not linearly changed but a sudden change as X larger than 0.61?

Response:

Thanks for this comment!

To determine whether a new phase is present between $x = 0.60$ and $x = 0.75$, we

increased the sample size and synthesized the series $(S\text{-3AMP})_x(R\text{-3AMP})_{1-x}$ ($x = 0.5, 0.55, 0.60, 0.65, 0.70, 0.75, 0.80, 0.85, 0.90, 0.95, \text{ and } 1.0$) by solid phase synthesis method. The main text is revised as:

PXRD shows that there are both **1*Rac*** and **1*S*** structural features when x is 0.65 and 0.70, which may be an intermediate phase or a mixture of the two phases (Fig. 3a, and Supplementary Fig. 10, 11a, 11b). To exclude the possibility of being a physical mixture, DSC and dielectric constant measurements were performed. As shown in Supplementary Fig. 11c and 11d, the curves of the physical mixture indicate the superimposed phase transition characteristics of both **1*Rac*** and **1*S***. In contrast, the samples with x 0.65 and 0.70 obtained by the SPS show characteristics of a single phase, rather than a mixture (Fig. 3b, and Supplementary Fig. 12d, 12e). So, we suppose that these two ratios may be intermediate phases. (Supplementary Fig. 10 see Page 7, Supplementary Fig. 11 and 12 see Page 8)

Fig. 3 | Structures and phase transition properties of $(S\text{-3AMP})_x(R\text{-3AMP})_{1-x}\text{PbBr}_4$. **a** PXRD patterns; **b** DSC curves; **c** Nonlinear correlation between the T_C and x (0.5 – 1). **d** Curves of enthalpy change, entropy change, and N with x . **e** Phase diagram and the point groups. The orange region is roughly estimated, due to the difficulty in obtaining ideal crystals. **f** Linear correlation between the T_C and x (0.70 – 1). The solid lines fit a linear regression model with the corresponding R^2 value, 95% confidence band, and 95% prediction band (shaded region).

3. The phase transition temperature's linear tuning is well-established. Specifically, is there a linear shift in the Flack parameter, ranging from 0 for chiral compounds to 0.5 for racemate compounds?

Response:

We cannot get single crystals with good enough quality for structural analysis. We expect to solve this problem in future work.

4. Which physical property can be linearly tuned and which is not with this method?

Response:

According to the present experimental results, we have shown that the phase transition temperature, enthalpy change, dielectric, SHG intensity, PL intensity, and chiral properties of the material can be regulated by this method. We also suppose the parameters such as coercive field, residual polarization value, and saturation polarization value can be further tuned. However, due to the lack of single crystals, we have not been able to measure them.

5. It indicates that the spacing of adjacent [PbBr₄] inorganic layers decreases from 10.09 Å of 1S to 10.04 Å of 1Rac (Figure 2b). In this sentence, Figure 2b does not contain the values of 10.09 and 10.04. Please check the figure citation.

Response:

The sentence was revised as:

According to Bragg's equation $2d\sin\theta = n\lambda$, the corresponding spacing d of adjacent [PbBr₄] inorganic layers decreases from 10.09 Å (**1S**) to 10.04 Å (**1Rac**), consistent with the ones shown in the single crystal structures (Fig. 2c, 2e).

6. What is the definition of angle beta in Figure 2? Are they the in-plane angle and out-of-plane angle? Please refer to Chem. Sci., 2017, 8, 4497

Response:

The β angle in Figure 2 represents the deviation of $\angle\text{Pb-Br-Pb}$ from the standard 180° . We do not distinguish between inner and outer plane angles, which is different from the situation in the literature (*Chem. Sci.*, 2017, 8, 4497).

7. In Figure 2c-d, a lot of data is not mentioned in the main text. Please give more details about them.

Response:

More details are added in the “Structural analysis and comparison of *1R/S* and *1Rac*” section in the main text:

According to Bragg's equation $2d\sin\theta = n\lambda$, the corresponding spacing d of adjacent $[\text{PbBr}_4]$ inorganic layers decreases from 10.09 \AA (**1S**) to 10.04 \AA (**1Rac**), consistent with the ones shown in the single crystal structures (Fig. 2c, 2e). However, this result seems counterintuitive, that is, the larger the d value, the weaker the interaction between the cation and the inorganic anion and so the lower the T_C . To figure out this point, we further studied the single crystal structures and found that the N–H \cdots Br distances in **1S** ($2.50 - 2.72 \text{ \AA}$) are shorter than the corresponding ones in **1Rac** ($2.55 \text{ \AA} - 2.91 \text{ \AA}$) (Fig. 2c, 2e), meaning stronger N–H \cdots Br interactions. This causes the lengthened distance of adjacent axial Br-defined planes from 3.78 \AA in **1S** to 3.92 \AA in **1Rac**, and the shortened distance of adjacent axial Pb-defined planes from 10.09 \AA in **1S** to 10.04 \AA in **1Rac**, (Fig. 2c, 2e). Therefore, the $[\text{PbBr}_6]$ octahedron in **1S** produces more distortion vertically than **1Rac** ($163.1^\circ < 170.5^\circ$) (Fig. 2c, 2e), and greater inter-octahedral compression horizontally than **1Rac** ($170.7^\circ < 175.2^\circ$) (Fig. 2d, 2f), which further reduces the space for the cation to move and increases the rotational barrier. From the above analysis, the molecular chirality can significantly affect the octahedral distortions and local molecular packings, and thus the T_C of the crystals.

8. It is better to present the interesting results by drawing a phase transition temperature change diagram with *R-rac-S* sequence as X changes.

Response:

Thanks for this valuable suggestion!

We agree with the reviewer that plotting the phase transition temperature as a function of the enantiomer fraction (0 – 1) can better present the results. Therefore, we added experiments and updated the phase diagram (Fig 3e)

Fig. 3e | Phase diagram and the point groups. The orange region is roughly estimated, due to the difficulty in obtaining ideal crystals.

9. Can it get similar results by mechanically grinding the mixture of pure *R* and *S* compounds?

Response:

Similar results can be obtained by mechanically grinding the mixture of pure *R* and *S* compounds. For example, the mixture of equal amounts of **1*R*** and **1*S*** can obtain **1*Rac*** (1:1) after mechanical grinding, whose phase purity is verified by PXRD, and the phase transition properties are further verified by DSC.

Note: to avoid confusion, we adopted solid-phase syntheses as the standard method to prepare the series since the *x* values can be accurately determined while the solution method does not and needs further characterizations.

10. The grammar of the manuscript needs to be improved? Such as: Crystalline powdered samples of **1*Rac*** show CPLEPL responses of about 8% for the L/R-CPL excitation at 395 nm and 298 K (Figure 5b), which is even comparable with those

of R/S-(BPEA)₂PbI₄. It's better to revise them as follows: Crystalline powder sample of 1Rac shows a CPLEPL response difference of about 8% upon the L/R-CPL excitation xxxx.

Response:

Thanks!

We revised these errors.

11. Figure S7, S9, S11, and S12 are not mentioned in the main text. Please mark their position in the main text.

Response:

We updated the figures and marked them properly in the main text.

12. In addition to the dielectric, piezoelectric, pyroelectric, and nonlinear optical effects, the multiaxial feature stands out as another crucial property for assessing ferroelectricity. The compound mentioned in the report crystallizes in the Cc space group. However, the polarization direction is not specified in measuring the P-E loop. Please refer to a similar perovskite ferroelectric compound (Natl Sci Rev. 2020 Sep 8;8(5): nwa232) with the sample space group.

Response:

According to the references, the direction of polarization is indicated in the paper, and the relevant literature is cited. The sentence was revised as:

Direct evidence of the ferroelectricity was confirmed by the well-shaped electric hysteresis loop by using the double-wave method performed on the crystal along the *c*-axis (Fig. 1e).⁵²

Reference:

52. Huang, C.-R.; Luo, X.; Chen, X.-G.; Song, X.-J.; Zhang, Z.-X.; Xiong, R.-G. A multiaxial lead-free two-dimensional organic-inorganic perovskite ferroelectric. *Natl. Sci. Rev.* **8**, nwa232 (2021).

Reviewer #5 (Remarks to the Author):

Key results

In the paper submitted by Fan et al., it is reported that the ferroelectric transition temperature of hybrid halide perovskite compounds could be tuned by adjusting the enantiomer fraction x of the chiral organic cations. Additionally, the experimental results demonstrate distinct circularly polarized light-excited photoluminescence (CPLEPL) responses (P) for compounds with different values of x .

Based on these experimental findings, the authors assert two main points:

1. A clear correlation was established between the enantiomer fraction x and the phase transition temperature T_c or P , with both exhibiting a linear relationship.
2. This novel "enantiomer fraction engineering" presents a facile and effective approach to investigating the structure and physical properties of chiral and polar perovskite compound assemblies.

Validity

Overview: The idea adopted as a strategy exhibits a certain degree of novelty and intrigue. While there are significant flaws in the experimental sample identification, the measurement results are clear. However, the analysis of the results is insufficient.

Response:

Thanks for this comment!

We added new experiments and made a more in-depth analysis of the experimental results.

- 1 The experimental results concerning the crystal structures, ferroelectric properties, phase transition temperatures, and CPLEPL for both 1R/S (ref.50), previously reported by the authors, composed of enantiopure cations and 1Rac composed of racemic cations have been adequately described.

Response:

Thanks for this comment!

2 The most crucial experimental concern resides in the absence of a methodological exposition regarding the determination of the enantiomer fraction x of the obtained samples. Careful verification should be done to ascertain whether the fraction attained in the obtained samples aligns with the mixing ratios set during synthesis. It is important to elucidate how the value of x was determined. (The ensuing comments are all made under the assumption that the provided values of x are accurate.)

Response:

Thanks for this valuable comment!

The mechanochemical method, also known as solid-phase synthesis (SPS), can better control the stoichiometric ratio of the mixed components than the solution method.^{54,55} To verify the effectiveness of this method, **1*Rac*** was prepared by equal molar amounts of **1*S*** and **1*R*** crystals. PXRD data show that the sample obtained by this method shows completely coincident peaks with the one obtained by the solution method using the racemic cation as a reactant (Supplementary Fig. 9a-c). So are the DSC results (Supplementary Fig. 9d). A series of $(S\text{-}3\text{AMP})_x(R\text{-}3\text{AMP})_{1-x}\text{PbBr}_4$ ($x = 0 - 1$ with an interval of every 0.05) was then synthesized by the SPS which ensures that the x value is exactly the mixing ratio set during the synthesis. (Supplementary Fig. 9 see Page 2)

3 Regarding the claim "Line 57: the phase transitions, \sim , showing a linear relationship between x and T_c or P ." The experimental results indeed demonstrate different T_c and P values for each sample obtained through mixing. However, the proposed linear relationship by the authors between x and T_c , as presented in the form of $T_c = 295 + 120x$ (Figure 3(h)), lacks clarity in terms of the underlying analytical model and raises concerns about its validity. Furthermore, elucidation should be provided regarding the

physical meanings of the values 295 and 120 obtained as parameters in this equation.

Response:

To obtain a more definitive relationship between enantiomer fraction and phase transition temperature, we increased the number of samples to be able to capture small changes in structural and property changes. The main text was revised as:

From Fig. 3b and Supplementary Fig. S12, it can be seen that the T_C varies with the change of enantiomer fraction. We select the DSC data in the heating process to draw Fig. 3c. When x increases from 0.5 to 0.65, the T_C decreases from 359 K to 355 K; and when x increases from 0.65 to 1, the T_C becomes to increase from 355 K to 432 K (a change of 77 K). At the same time, by following the Boltzmann equation, $\Delta S = R \ln(N)$, the enthalpy change ΔH , entropy change ΔS , and orientation number N with x also show similar trends to the T_C (Supplementary Table 4, and Fig. 3d). When $x = 0.70$, ΔH , ΔS , and N have the minimum values, which are 4.4 kJ/mol, 12.2 J/(mol·K), 4.34, respectively, which are characteristic of an order-disorder type of phase transitions. Therefore, we think that the enantiomer fraction does not change the phase transition type of the series. Furthermore, to verify the above experimental results, we synthesized samples with the x between 0 and 0.5 (Supplementary Fig. 13). By extracting the DSC data in the heating process, it is found that the T_C shows the same trend as $x = 0.5 - 1$ (Supplementary Fig. 13d).

This fully proves the effectiveness of this strategy.

Fig 3. | Structure and phase transition properties of $(S\text{-}3\text{AMP})_x(R\text{-}3\text{AMP})_{1-x}\text{PbBr}_4$.

4. Additionally, a linear relationship between x and P cannot be found from the presented experimental results.

Response:

The point was clarified in the main text:

By analyzing the P values of all samples, we find that the relationship between the x and P is similar to the trend of the SHG intensity, that is, the P value decreases when x increases from 0.5 to 0.65 and then increases when x from 0.65 to 1.0 (Fig. 5f and Supplementary Fig. 17 and 14c). There is a good linear correlation between x (0.65 – 1) and P (Fig. 5g).

CPLEPL spectra suggest that the enantiomeric excess can effectively regulate the P value.

Fig. 5 | f. Comparison of the P values of the perovskites with different enantiomer fractions. **g** Linear correlation between the P and x . The solid lines fit a linear regression model with the corresponding R^2 value, 95% confidence band, and 95% prediction band (shaded region).

Significance

1. There have been many reports discussing the examination of solids (= racemic solid solutions) obtained at various enantiomer fractions x . (For instance, J. Am. Chem. Soc. 2006, 128, 11985-11992, Crystal Growth & Design 2010, 10, 1808–1812.) Therefore, the novelty of the sample preparation method itself does not merit the proposition of a new nomenclature as a strategic innovation.

Response:

Chiral compounds have attracted attention, especially from the materials science field. For ferroelectrics, chirality is an efficient way to introduce ferroelectricity and new functions. We focus on the development and research of new ferroelectric phase change materials, but find there is still no research on the continuous regulation of material phase change temperature by enantiomer fraction. Therefore, we adopt the phrase “enantiomer fraction engineering” to describe its effectiveness in the structure and property regulations of chiral materials. It could attract the attention of peers in related fields and highlight the key of this method.

2. The investigation of the changes in physical properties (phase transition temperatures)

resulting from incremental variations in x has primarily revolved around melting points until now. The application of this approach to the ferroelectric phase transition phenomenon and the experimental demonstration of actual changes in TC offer novel contributions. Unfortunately, the authors have yet to present the material science significance or fundamental scientific insights into the ability to modulate phase transition temperatures "between racemic and R/S materials" using this approach. A more thorough analysis revealing similarities and discrepancies compared to the prior discussions on melting points would yield intriguing results.

Response:

We revised related descriptions in the Introduction section:

Retrospectively, research on enantiomer solid solutions mostly focused on the difference in solubility and melting point of compounds with different enantiomer fractions of organic molecular crystals, which is generally used to guide chiral separation.^{48,49} In organic chemistry, how to get enantiomerically pure molecules is always the focus because compounds with different enantiomeric purities often show different biological activities.^{50,51} Generally, the potentials of materials with different enantiomer fractions have been overlooked in the discovery of new structures and properties as well as the cost consideration in material processing.

3. As the authors suggest, the possibility of similar modulation in analogous compounds, as summarized in Figure 4, seems plausible. In the stage of implementing this in practical applications within society, the potential for such fine-tuning may be important.

Response:

Thanks!

Data and methodology

1. Figure 1 depicts several characteristics supporting a phase transition occurring at 359 K: (a) a rapid change in dielectric permittivity, (b) a peak in differential scanning calorimetry (DSC), (c) order-disorder transitions of cation molecules in the crystal

structure, (d) a sudden rise in second harmonic generation (SHG), and (e) hysteresis in the polarization-electric field (P-E) curve. I see no major problems with these data.

2. In Figure 2, it becomes evident from the DSC peak at different temperatures (a) that 1S and 1Rac exhibit distinct phase transition temperatures. Although the peak areas appear significantly disparate, questions arise regarding whether the transition mechanisms could be regarded as identical for both. While the difference in rotational ease of the 3-AMP cation is attributed to the distortion of the PdBr6 octahedral structure and N-H-Br bond distances in single-crystal X-ray diffraction (SCXRD), offering a plausible explanation for the T_c disparity, it is also conceivable to discuss the same issue by comparing the volumes of the spaces occupied by the cations. Additionally, the variation in distortion of the PdBr6 octahedral structure presented here could potentially provide insights into the disparity in polarization values (0.50 uCcm^{-2} for 1Rac and 1.0 uCcm^{-2} for 1S/R).

Response:

Thanks for this comment!

At the same time, by following the Boltzmann equation, $\Delta S = R \ln(N)$, the enthalpy change ΔH , entropy change ΔS , and orientation number N with x also show similar trends to the T_c (Supplementary Table 4, and Fig. 3d). When $x = 0.70$, ΔH , ΔS , and N have the minimum values, which are 4.4 kJ/mol , $12.2 \text{ J/(mol}\cdot\text{K)}$, 4.34 , respectively, which are characteristic of an order-disorder type of phase transitions. Therefore, we think that the enantiomer fraction does not change the phase transition type of the series.

Supplementary Table 4. Summary of phase transition temperature (T_c), enthalpy change (ΔH), entropy change (ΔS), and orientation number (N) of $(S\text{-}3\text{AMP})_x(\text{R}\text{-}3\text{AMP})_{1-x}\text{PbBr}_4$ ($x = 0.5 - 1$).

EFE	T_{Heating}	ΔH	ΔH	ΔS^a	N^b
0.50	359.1 K	4.9 J/g	6.3 kJ/mol	17.5 J/(mol·K)	8.21
0.55	359.4 K	4.9 J/g	6.3 kJ/mol	17.5 J/(mol·K)	8.21
0.60	356.4 K	4.8 J/g	6.2 kJ/mol	17.4 J/(mol·K)	8.11

0.65	355.3 K	3.9 J/g	5.0 kJ/mol	14.1 J/(mol·K)	5.45
0.70	359.4 K	3.4 J/g	4.4 kJ/mol	12.2 J/(mol·K)	4.34
0.75	370.5 K	4.7 J/g	6.0 kJ/mol	12.7 J/(mol·K)	4.61
0.80	383.2 K	6.0 J/g	7.7 kJ/mol	20.1 J/(mol·K)	11.22
0.85	393.5 K	7.3 J/g	9.4 kJ/mol	23.9 J/(mol·K)	17.72
0.90	405.5 K	8.0 J/g	10.3kJ/mol	25.4 J/(mol·K)	21.22
0.95	419.0 K	9.0 J/g	11.6 kJ/mol	27.7 J/(mol·K)	27.99
1.00	432.7 K	10.3 J/g	13.2 kJ/mol	30.1 J/(mol·K)	37.35

Fig. 3d | Curves of enthalpy change, entropy change, and N with x

3. In Figure 3, PXRD patterns (a) and (d) suggest crystallization in space group Cc for $x = 0.5$ and 0.61 , while $x = 0.75-1.0$ implies crystallization in space group $P21$. This indicates the formation of racemic solid solutions rather than mixtures of racemic and enantiopure crystals. The difference in phase transition temperatures can be confirmed by the DSC comparison (c). However, the lack of displayed vertical axis values prevents a comparison of peak intensities, thereby withholding essential information concerning the mechanism of the phase transition. Schematic diagram (g) for $x = 0.75$ gives the impression of highly ordered substitution to the R configuration within a crystal. In reality, wouldn't substitution within the solid solution occur randomly?

Response:

Thanks for this comment!

The diagram is revised as follows:

4. In Figure 5 (b)-(g), the alteration of CPLEPL spectra as x varies is notably clear.

Response:

Thanks for this comment.

5. Discrepancies arise between the experimental method outlined in the Supporting Information for single-crystal structure analysis and the contents of the submitted CIF file.

Response:

We have revised the description of the single-crystal structure analysis experimental method, as shown below:

Rigaku CrysAlisPro software was used to collect data, refine cell, and reduce data. SHELXL-2018 with the OLEX2 interface was used to solve the structures by direct methods.

6. Additionally, discrepancies also exist between the values in Table S1 and the content of the CIF file, particularly in parameters such as the Flack parameter and R1/wR2 values. These issues necessitate rectification.

Response:

We updated the structure in the CCDC database and corrected the crystallographic data

in Supplementary Table 1.

Analytical approach

5.1 There seems to be room for discussion regarding the linear analysis in Figure 3(h).

The PXRD results identify structural changes between Cc–C2/c for low fraction samples and between P21–P422 for high fraction samples. It does not seem appropriate to evaluate the phase transition temperatures (which are related to ΔH and ΔS) of such a system with different symmetry changes using a simple linear correlation together.

Response:

These issues have been addressed and are explained in detail in the **Suggested Improvements** section below. (see page 47)

5.2 Figure 5(h) appears to lack any analysis or trends beyond the observed change in P with different enantiomer fractions x .

Response:

We analyzed it in the main text:

By analyzing the P values of all samples, we find that the relationship between the x and P is similar to the trend of the SHG intensity, that is, the P value decreases when x increases from 0.5 to 0.65 and then increases when x from 0.65 to 1.0 (Fig. 5f and Supplementary Fig. 17 and 14c). There is a good linear correlation between x (0.65 – 1) and P (Fig. 5g). Both the P value and the SHG strength are properties that are related to the non-centrosymmetric properties of the materials.^{27,67-68} So we speculate that their microscopic origins can be ascribed to the changes of the non-centrosymmetry by enantiomer fraction-induced distortions of the inorganic sublattice (Fig. 3g and Supplementary Fig. 14c).

Supplementary Fig. 17d. | Studies of Circularly polarization-sensitive PL spectroscopy.

Supplementary Fig. 14c | Comparison of the P values of the perovskites with different enantiomer fractions.

Fig. 5g | Linear correlation between the P and x . The solid lines fit a linear regression model with the corresponding R^2 value, 95% confidence band, and 95% prediction band (shaded region).

Fig. 3g | PXRD peak representing the $[\text{PbBr}_4]^{2-}$ inorganic plane of $(S\text{-}3\text{AMP})_x(R\text{-}3\text{AMP})_{1-x}\text{PbBr}_4$ ($x = 0 - 0.5$)

Suggested improvements

1. The actual determination of enantiomer fractions in the obtained racemic solid

solutions is necessary. This could be achieved by redissolving the obtained solids, followed by extraction of the organic material and subsequent analysis using techniques such as chiral HPLC.

Response:

Thanks for this valuable suggestion!

Since the proportion of products obtained by chemical synthesis is not necessarily equal to the proportion of raw materials input, we considered using other methods to solve this problem, such as trying to redissolve the obtained solids, then extract the organic matter and subsequently confirm the analysis using techniques such as chiral high-performance liquid chromatography, or by directly measuring the optical rotation of perovskite solution. However, two-step synthesis, extraction, and purification will greatly affect the accuracy of the data. At the same time, because the angle of rotation of the enantiomer pure molecule is very small, it will affect the determination of perovskite with a low enantiomer fraction.

The corresponding main text was revised as:

The mechanochemical method, also known as solid-phase synthesis (SPS), can better control the stoichiometric ratio of the mixed components than the solution method.^{54,55}

To verify the effectiveness of this method, **1*Rac*** was prepared by equal molar amounts of **1*S*** and **1*R*** crystals. PXRD data show that the sample obtained by this method shows completely coincident peaks with the one obtained by the solution method using the racemic cation as a reactant (Supplementary Fig. 9a-c, page 22). So are the DSC results (Supplementary Fig. 9d, page 22).

Subsequent experimental characterizations were performed based on the newly synthesized sample.

2. The peaks observed in DSC measurements should provide information about the entropy change (ΔH) associated with the phase transition. Given its significance when dealing with phase transition phenomena, a comparative analysis is crucial. Additionally, calculating the entropy (ΔS) might help assess whether the phase transition is truly order-disorder or displacement type.

Response:

The samples synthesized by the solid phase synthesis method were tested by DSC.

The main text was shown as:

At the same time, by following the Boltzmann equation, $\Delta S = R \ln(N)$, the enthalpy change ΔH , entropy change ΔS , and orientation number N with x also show similar trends to the T_c (Supplementary Table 4, and Fig. 3d). When $x = 0.70$, ΔH , ΔS , and N have the minimum values, which are 4.4 kJ/mol, 12.2 J/(mol·K), 4.34, respectively, which are characteristic of an order-disorder type of phase transitions. Therefore, we think that the enantiomer fraction does not change the phase transition type of the series. (Supplementary Table 4 see page 43, and Fig. 3d see page44)

3. An explanation is required for the observed changes in circularly polarized luminescence emission (CPLEPL) when P varies with different enantiomer fractions. Moreover, considering the definition of P , it seems to relate to enantiomeric excess (e.e.) rather than enantiomer fractions x . Plotting the x-axis of Figure 5(h) in terms of e.e. might aid in interpreting the results.

Response:

We added an explanation.

Meanwhile, as shown in the figure below, we establish a relationship between P and enantiomer excess (e.e.), but it shows the same trend and relationship as enantiomer fraction and p -value. Enantiomer excess (e.e.) is often used to indicate the enantiomer purity of a compound. However, in our system, enantiomer excess is not as good as enantiomer fraction to directly represent enantiomer content in a structure (Environ. Sci. Technol., 2020, 34, 218-220). So we would like to use enantiomer fractions to analyze the relationship between them and their properties.

Clarity and context

1. I find it challenging to discern the clear purpose behind the authors' proposed methodology for tuning the ferroelectric phase transition temperature. Whether it aims to introduce a new fundamental scientific strategy, highlight novel phenomena, enhance material properties from a materials science perspective, or reduce environmental impact as part of green chemistry remains unclear. In the introduction, the authors criticize the approach of prior studies that observed changes in T_c , but the content of their critique lacks consistency. This has created ambiguity in the authors' stance towards this study.

Response:

In this paper, we not only want to highlight this new strategy from the perspective of improving material properties but also want to call for increasing the utilization rate of chiral molecules from environmental and economic perspectives. For a clear description, we revised the corresponding part in the Introduction section:

In molecular ferroelectric crystals, the shape and size of the molecules greatly regulate the packing structures, which makes it possible to optimize the ferroelectricity through rational molecular design.⁴³ However, due to the highly sensitive nature of ferroelectricity to molecular structures and intermolecular interactions, sometimes, even substituting a single atom can result in the disappearance of ferroelectricity. Chirality has the advantage of having nearly the same intermolecular interactions with neighbors in the crystal structures, which can allow finely tuning of the T_c by

maintaining the ferroelectricity. Moreover, as a basic molecular structure feature, chirality has been exploited to design and construct chiral&ferroelectric materials to achieve bulk photovoltaic effect,⁴⁴ chiroptical-coupled ferroelectric properties such as circularly polarized luminescence,^{45,46} and circularly polarized light (CPL) detection.⁴⁷ However, in these studies, little attention has been paid to the impact of enantiomeric excess on the structures and properties of the ferroelectrics, and the use of molecules with different chiral purities has never been considered when developing new materials. Retrospectively, research on enantiomer solid solutions mostly focused on the difference in solubility and melting point of compounds with different enantiomer fractions of organic molecular crystals, which is generally used to guide chiral separation.^{48,49} In organic chemistry, how to get enantiomerically pure molecules is always the focus because compounds with different enantiomeric purities often show different biological activities.^{50,51} Generally, the potentials of materials with different enantiomer fractions have been overlooked in the discovery of new structures and properties as well as the cost consideration in material processing.

2. The presentation of experimental results has generally been executed without significant issues. However, I find the descriptions regarding the interpretation and analysis of the results to be insufficient. Consequently, assertions such as the establishment of a "clear correlation between enantiomer fraction and phase transition temperature" and the "novelty and utility of the enantiomer fraction engineering approach" were difficult to interpret as having sufficient accuracy. I believe that a careful comparison with prior research results on the melting points of racemic solid solutions and a meticulous thermodynamic interpretation of DSC data would greatly enhance the quality of this study.

Response:

We increase the sample size and synthesize the series with an interval of every 0.05 by solid-phase synthesis method. Based on DSC data, the enthalpy changes of the materials were analyzed.

The main text was revised as:

At the same time, by following the Boltzmann equation, $\Delta S = R \ln(N)$, the enthalpy change ΔH , entropy change ΔS , and orientation number N with x also show similar trends to the T_C (Supplementary Table 4, and Fig. 3d). When $x = 0.70$, ΔH , ΔS , and N have the minimum values, which are 4.4 kJ/mol, 12.2 J/(mol·K), 4.34, respectively, which are characteristic of an order-disorder type of phase transitions. Therefore, we think that the enantiomer fraction does not change the phase transition type of the series. (Supplementary Table 4 see page 43, and Fig. 3d see page44)

3. The results on the enantiomer fraction x dependency of CPLEPL are presented in conjunction. Nevertheless, their relevance to the title's assertion of "tuning the ferroelectric phase transition temperature" remains unclear.

Response:

By analyzing the P value of the samples (Supplementary Fig. 17), we find that the relationship between x and P is the same as the relationship between x and the phase transition temperature and SHG intensity. Combining with the above PXRD analysis, we can see that the change of enantiomer fraction can affect the noncentral symmetry of the structure by inducing the distortion of the inorganic sublattice (Fig. 3g). Both the P value and the SHG strength are properties that are closely related to the non-centrosymmetric properties of the materials, so we speculate that this is the main microstructure source of P-value, phase transition temperature and SHG intensity changes.

4. Regarding terminology:

In L96, "the chiral & polar space group P21 (point group 2)" and L99, "chiral space group P422" are mentioned, yet P21 and P422 are not considered chiral space groups. Please verify the 22 unique chiral space groups (11 pairs) as listed in references such as J. Appl. Cryst. (2018). 51, 1481–1491.

Response:

Thanks!

The sentences were revised as:

Both **1R** and **1S** crystallize in the space group $P2_1$ (polar point group 2) in the FEP (Supplementary Fig. 8a).

The PEP belongs to the SHG-inactive space group $P422$ (point group 422) (Supplementary Fig. 8e).

The use of "homochiral" to convey enantiopure is not recommended by IUPAC and should be avoided. Exercise caution when employing "homochiral" concerning molecules. For reference, consult the IUPAC Gold Book entry on "enantiomerically pure (enantiopure)."

Response:

Thanks!

We have checked the full text and revised the corresponding descriptions.

References

1. Unfortunately, the cited papers are somewhat inappropriate. Relying solely on examples from the domain of halide perovskites under the author's investigation when discussing fundamentally rooted and general phenomena such as phase transitions and well-established areas like the properties of ferroelectric materials is inadequate.

Response:

When we discuss the properties of phase transformation and ferroelectric materials, we cite studies in fields including halide perovskite, perovskite-type oxides, metal complexes, single-component organic crystals, polymers, fresnoite crystals, two-dimensional materials, molecular crystals, and specific citations are as follows:

References:

1. Li, M.; Han, S.; Liu, Y.; Luo, J.; Hong, M.; Sun, Z. Soft perovskite-type antiferroelectric with giant electrocaloric strength near room temperature. *J. Am. Chem. Soc.* **142**, 20744–20751 (2020).
3. Jiang, C.; Zhang, C.; Li, F.; Sun, L.; Li, Y.; Yu, F.; Zhao, X. Phase transition regulation and piezoelectric performance optimization of fresnoite crystals for high-temperature acceleration sensing. *J. Mater. Chem. C* **10**, 180-190 (2022).

4. Li, Y.; Wu, S.-Q.; Xue, J.-P.; Wang, X.-L.; Sato, O.; Yao, Z.-S.; Tao, J. A molecular crystal shows multiple correlated magnetic and ferroelectric switchings. *CCS Chem.* **3**, 2464-2472 (2021).
5. Sun, Z.; Tang, Y.; Zhang, S.; Ji, C.; Chen, T.; Luo, J. Ultrahigh pyroelectric figures of merit associated with distinct bistable dielectric phase transition in a new molecular compound: Di-n-butylammonium trifluoroacetate. *Adv. Mater.* **27**, 4795-4801 (2015).
6. Yao, W.-W.; Ma, B.-B.; Xu, L.; Shao, D.-S.; Qian, Y.; Liu, W.-L.; Ren, X.-M. Structural, magnetic and phase transition properties in $s = \frac{1}{2}$ radical solid solutions of $[f_xcl_{1-x}bzpy][ni(mnt)_2]$ ($x = 0.07-0.87$). *Inorg. Chem. Front.* **10**, 2325-2334 (2023).
7. Lu, H.; Bark, C. W.; Esque de los Ojos, D.; Alcalá, J.; Eom, C. B.; Catalan, G.; Gruverman, A. Mechanical writing of ferroelectric polarization. *Science* **336**, 59-61 (2012).
8. Zhang, H.-Y.; Jiang, H.-H.; Zhang, Y.; Zhang, N.; Xiong, R.-G. Ferroelectric lithography in single-component organic enantiomorphic ferroelectrics. *Angew. Chem. Int. Ed.* **61**, e202200135 (2022).
9. Liu, Z.; Hou, P.; Sun, L.; Tsymbal, E. Y.; Jiang, J.; Yang, Q. In-plane ferroelectric tunnel junctions based on 2d α -in2se3/semiconductor heterostructures. *npj Comput. Mater.* **9**, 6 (2023).
10. Liu, Z.; Wang, H.; Li, M.; Tao, L.; Paudel, T. R.; Yu, H.; Wang, Y.; Hong, S.; Zhang, M.; Ren, Z.; Xie, Y.; Tsymbal, E. Y.; Chen, J.; Zhang, Z.; Tian, H. In-plane charged domain walls with memristive behaviour in a ferroelectric film. *Nature* **613**, 656-661 (2023).
13. Neese, B.; Chu, B.; Lu, S.-G.; Wang, Y.; Furman, E.; Zhang, Q. M. Large electrocaloric effect in ferroelectric polymers near room temperature. *Science* **321**, 821-823 (2008).
15. Horiuchi, S.; Kagawa, F.; Hatahara, K.; Kobayashi, K.; Kumai, R.; Murakami, Y.; Tokura, Y. Above-room-temperature ferroelectricity and antiferroelectricity in benzimidazoles. *Nat. Commun.* **3**, 1308 (2012).

2. Moreover, chirality has been well studied not only concerning enantioselective synthesis or pharmaceutical bioactivity but also more fundamental physical properties. The citation of references 48 and 49 in this context proves difficult to comprehend. Notably absent is the consideration of prior research on phase transitions in chiral solid solutions.

Response:

References 48 and 49 were renumbered as references 50 and 51.

We revised this part in the introduction section:

Retrospectively, research on enantiomer solid solutions mostly focused on the difference in solubility and melting point of compounds with different enantiomer

fractions of organic molecular crystals, which is generally used to guide chiral separation.^{48,49} In organic chemistry, how to get enantiomerically pure molecules is always the focus because compounds with different enantiomeric purities often show different biological activities.^{50,51} Generally, the potentials of materials with different enantiomer fractions have been overlooked in the discovery of new structures and properties as well as the cost consideration in material processing.

References:

48. Li, W.; de Groen, M.; Kramer, H. J. M.; de Gelder, R.; Tinnemans, P.; Meekes, H.; ter Horst, J. H. Screening approach for identifying cocrystal types and resolution opportunities in complex chiral multicomponent systems. *Cryst. Growth Des.* **21**, 112-124 (2020).
49. Huang, J.; Chen, S.; Guzei, I. A.; Yu, L. Discovery of a solid solution of enantiomers in a racemate-forming system by seeding. *J. Am. Chem. Soc.* **128**, 11985-11992 (2006).
50. Ali, I.; Singh, P.; Aboul-Enein, H. Y.; Sharma, B. Chiral analysis of ibuprofen residues in water and sediment. *Anal. Lett.* **42**, 1747-1760 (2009).
51. Izake, E. L. Chiral discrimination and enantioselective analysis of drugs: An overview. *J. Pharm. Sci.* **96**, 1659-1676 (2007).

My Expertise

I lack the expertise to evaluate the analysis and assessment of semiconductor properties through DFT calculations, as well as the interpretation of the computational methodology and its outcomes.

REVIEWER COMMENTS

Reviewer #1 (Remarks to the Author):

The authors have carefully considered all remarks. The revised version of the manuscript is recommended for publication in its present form.

Reviewer #2 (Remarks to the Author):

In this round revision, the authors have made detailed responses to all the reviewers' comments and suggestions. The manuscript is well improved in terms of the revised contents, which can be accepted for publication now.

Reviewer #3 (Remarks to the Author):

accepted as it is.

Reviewer #5 (Remarks to the Author):

In the revised manuscript, the results of experiments with an increased sample size and their thermodynamic analysis have been incorporated. This addition enhances the clarity of the authors' assertion regarding the "Adjustment of Ferroelectric Phase Transition Temperature by Enantiomer Fraction." Most issues raised in previous comments have been addressed. The following are comments on the current manuscript.

1. The determination of enantiomeric fractions (EF) for each sample, which was a significant concern in the prior drafts, has not been executed.

It may be accepted that the conversion of synthetic methods to solid-phase reactions improves the credibility of the EFs.

However, unless experimentally and directly determined, these values merely represent anticipated EF values and cannot be conclusively recognized as the EFs for the respective samples. Therefore, the assertion in line 135, "synthesized by the SPS which ensures that the x value is exactly the mixing ratio set during the synthesis," is, at least for my part, untenable.

In consideration of the aforementioned issues, the following two modifications are proposed:

1-a Explicitly state around lines 136 and 255 that "the determination of enantiomeric fraction could not be determined for the prepared samples."

1-b Change the horizontal axis label "Enantiomer Fraction (x)" utilized in the graph as either "Expected Enantiomer Fraction (x)" or "Mixing Ratio (x)."

2. Please provide the equation used for the fitting curve of the "Nonlinear correlation" in Figure 3(c), along with the fitting parameters.

3. Figure 3(e) presents a notably interesting and compelling illustration. However, I find it difficult to comprehend the significance of extracting a portion of it in Figure 3(f) to emphasize the "linear relationship." The main text lacks an explanation of the models or meanings associated with establishing a linear relationship.

4. EFs are just values ranging from 0.5 to 1.0. The question remains whether simply changing the value can be called "engineering." For example, "Different concentrations of saline solutions were prepared in the range of 0-26% and their freezing points were different (or tuned)." Referring to such phenomena as "concentration engineering" induces a sense of discomfort within me. Perhaps a more natural title could be "Tuning Ferroelectric Phase Transition Temperature by Enantiomer Fraction"

RESPONSE TO REVIEWERS' COMMENTS

Reviewer #1 (Remarks to the Author):

The authors have carefully considered all remarks. The revised version of the manuscript is recommended for publication in its present form.

Response:

Thanks for the comment!

Reviewer #2 (Remarks to the Author):

In this round revision, the authors have made detailed responses to all the reviewers' comments and suggestions. The manuscript is well improved in terms of the revised contents, which can be accepted for publication now.

Response:

Thanks for the comment!

Reviewer #3 (Remarks to the Author):

accepted as it is.

Response:

Thanks for the comment!

Reviewer #5 (Remarks to the Author):

In the revised manuscript, the results of experiments with an increased sample size and their thermodynamic analysis have been incorporated. This addition enhances the clarity of the authors' assertion regarding the "Adjustment of Ferroelectric Phase Transition Temperature by Enantiomer Fraction." Most issues raised in previous comments have been addressed.

Response:

Thanks for the comment!

The following are comments on the current manuscript.

1. The determination of enantiomeric fractions (EF) for each sample, which was a significant concern in the prior drafts, has not been executed. It may be accepted that the conversion of synthetic methods to solid-phase reactions improves the credibility of the EFs. However, unless experimentally and directly determined, these values merely represent anticipated EF values and cannot be conclusively recognized as the EFs for the respective samples. Therefore, the assertion in line 135, "synthesized by the SPS which ensures that the x value is exactly the mixing ratio set during the synthesis," is, at least for my part, untenable. In consideration of the aforementioned issues, the following two modifications are proposed:

1-a Explicitly state around lines 136 and 255 that "the determination of enantiomeric fraction could not be determined for the prepared samples."

Response:

Thanks for the comment!

The sentence was revised as: "A series of $(S\text{-}3\text{AMP})_x(R\text{-}3\text{AMP})_{1-x}\text{PbBr}_4$ ($x = 0 - 1$ with an interval of every 0.05) was then synthesized by the SPS which ensures that the x value is exactly the mixing ratio set during the synthesis (note: the enantiomeric fraction could not be determined experimentally for the prepared samples and the value is the expected enantiomer fraction)."

1-b. Change the horizontal axis label "Enantiomer Fraction (x)" utilized in the graph as either "Expected Enantiomer Fraction (x)" or "Mixing Ratio (x)."

Response:

The horizontal label "Enantiomer Fraction (x)" was changed to "Expected Enantiomer Fraction (x)" in the graphs both in the main text and SI.

2. Please provide the equation used for the fitting curve of the "Nonlinear correlation" in Figure 3(c), along with the fitting parameters.

Response:

The equations and fitting parameters used are provided as shown below:

Supplementary Table 5. Polynomial fitting equations and fitting parameters.

Equation	$y = \text{Intercept} + B_1 * x + B_2 * x^2$
Plot	T_C
Weight	No Weighting
Intercept	491.12622 ± 24.15133
B_1	-476.3049 ± 66.38703
B_2	420.97902 ± 44.06747
Residual Sum of Squares	83.30931
R-Square (COD)	0.98912
Adj. R-Square	0.9864

3. Figure 3(e) presents a notably interesting and compelling illustration. However, I feel it difficult to comprehend the significance of extracting a portion of it in Figure 3(f) to emphasize the "linear relationship." The main text lacks an explanation of the models or meanings associated with establishing a linear relationship.

Response:

Thanks for the comment!

For the "linear relationship" in Figure 3(f), we want to emphasize that continuous control of the phase transition temperature can be achieved within a certain interval through this strategy.

We revised the description of this part as:

More importantly, when x is in the range of 0.70 – 1, a strong linear correlation exists between the x and T_C (Fig. 3f). This ensures that the enantiomer fraction engineering

strategy, as a simple and novel method, can continuously and linearly tune the composition and properties of chiral materials

4. EFs are just values ranging from 0.5 to 1.0. The question remains whether simply changing the value can be called “engineering.” For example, “Different concentrations of saline solutions were prepared in the range of 0-26% and their freezing points were different (or tuned).” Referring to such phenomena as “concentration engineering” induces a sense of discomfort within me.

Perhaps a more natural title could be “Tuning Ferroelectric Phase Transition Temperature by Enantiomer Fraction”

Response:

Thanks for the comment!

The title was revised as: “Tuning Ferroelectric Phase Transition Temperature by Enantiomer Fraction”.

REVIEWERS' COMMENTS

Reviewer #5 (Remarks to the Author):

The issues I pointed out have been addressed in this revision.
I think that the description is now comprehensive enough for the reader to assess the content.

RESPONSE TO REVIEWERS' COMMENTS

Reviewer #5 (Remarks to the Author):

The issues I pointed out have been addressed in this revision.

I think that the description is now comprehensive enough for the reader to assess the content.

Response:

Thanks for the comment!